# Frequency and nature of potentially harmful preventable problems in primary care from the patient's perspective with clinician review: a population-level survey in Great Britain

Susan Jill Stocks,[1] Ailsa Donnelly,[2] Aneez Esmail,[3] Joanne Beresford,[2] Sarah Luty,[4] Richard Deacon,[5] Avril Danczak,[6] Nicola Mann,[2] David Townsend,[2] James Ashley,[7] Carolyn Gamble,[2] Paul Bowie,[8,9] Stephen M Campbell[3]

For numbered affiliations see end of article.

**Correspondence to**
Dr Susan Jill Stocks;
sjstocks@btinternet.com

## ABSTRACT

**Objectives** To estimate the frequency of patient-perceived potentially harmful problems occurring in primary care. To describe the type of problem, patient predictors of perceiving a problem, the primary care service involved, how the problem was discussed and patient suggestions as to how the problem might have been prevented. To describe clinician/public opinions regarding the likelihood that the patient-described scenario is potentially harmful.

**Design** Population-level survey.

**Setting** Great Britain.

**Participants** A nationally representative sample of 3975 members of the public aged ≥15 years interviewed during April 2016.

**Main outcome measures** Counts of patient-perceived potentially harmful problems in the last 12 months, descriptions of patient-described scenarios and review by clinicians/members of the public.

**Results** 3975 of 3996 participants in a nationally representative survey completed the relevant questions (99.5%). 300 (7.6%; 95% CI 6.7% to 8.4%) of respondents reported experiencing a potentially harmful preventable problem in primary care during the past 12 months and 145 (48%) discussed their concerns within primary care. This did not vary with age, gender or type of service used. A substantial minority (30%) of the patient-perceived problems occurred outside general practice, particularly the dental surgery, walk in clinic, out of hours care and pharmacy. Patients perceiving a potentially harmful preventable problem were eight times more likely to have 'no confidence and trust in primary care' compared with 'yes, definitely' (OR 7.9; 95% CI 5.9 to 10.7) but those who discussed their perceived-problem appeared to maintain higher trust and confidence. Generally, clinicians ranked the patient-described scenarios as unlikely to be potentially harmful.

**Conclusions** This study highlights the importance of actively soliciting patient's views about preventable harm in primary care as patients frequently perceive potentially harmful preventable problems and make useful suggestions for their prevention. Such engagement may also help to improve confidence and trust in primary care.

## Strengths and limitations of this study

► We used a questionnaire co-designed with members of the public to quantify and describe patient-perceived potentially harmful preventable problems in primary care.

► The survey population was drawn from randomly selected group of addresses to give a representative sample of the Great Britain population.

► The potentially harmful preventable problems were self-reported by the survey respondents but primary care clinicians and members of the public estimated the likelihood that, in their opinion, each patient-described scenario was a potentially harmful preventable problem.

## BACKGROUND

Patients and clinicians view safety differently; patients tend to consider both serious safety problems and lesser causes of distress as safety concerns.[1] Patients judge quality and safety of care in terms of the ongoing care they receive over time, whereas healthcare professionals may take the view that they provide high-quality healthcare occasionally punctuated by discrete safety incidents and adverse events.[2] Even so patients can report medical errors accurately[3 4] but they may have different priorities to professionals, for example, prioritising psychological and emotional harm over technical errors.[5] Given these differences the patient's approach to preventing safety problems may differ from clinicians, particularly if they believe clinicians to be responsible for the problem rather than the institutional system.[6 7] Patient safety in primary care is rarely evaluated from the patient's perspective,[8] whereas involving patients in identifying errors and reducing

harm is common in secondary care.[3 9–11] A more participatory role for patients is advocated as a way to improve safety[12] suggesting a need for patients and professionals to be cognisant of each other's expectations and understanding of safety.

Estimates of the frequency of patient safety problems in primary care are generally from the clinician's perspective and range from >1 to 24 per 100 consultations or record review.[13–15] Some studies have quantified patient safety problems in primary care from the patient's perspective[6 7 16–18] However, quantitative patient-reported data from the UK are sparse; this may be partly due to the lack of a valid and reliable instrument for measuring safety in primary care from the patient's perspective.[19] The National Reporting and Learning System (NRLS) in England and Wales is a voluntary reporting scheme for National Health Service (NHS) staff to report patient safety incidents. Less than 1% of reports originate from primary care,[20] probably reflecting under-reporting. Until recently, patients could not make reports directly to the NRLS.[21 22] A European survey in 2013 found that 43% of UK respondents felt that it was 'likely' that patients could be harmed by non-hospital healthcare and a recent survey of the UK public found that 21% of respondents reported experiencing a potentially harmful preventable problem in primary care within the past 12 months.[23 24] These surveys suggest large differences between patients and clinicians in their beliefs about potentially harmful problems in primary care, but this has not been examined at the population level. The Patient Reported Experiences and Outcomes of Safety in Primary Care (PREOS-PC) questionnaire has reported qualitatively on patient perceptions of safety in English general practices finding that patient recommendations for safer healthcare included improvements in patient-centred communication, continuity of care, timely appointments, technical quality of care, active monitoring, teamwork, health records and practice environment.[25 26]

We aimed to quantify and describe patient-perceived potentially harmful preventable problems occurring in UK primary care. We also wanted to explore the differences in opinion between primary care professionals and the public regarding the potential for harm in the patient-described scenarios. Our approach aimed to capture the true patient perspective through extensive public and patient involvement (PPI); the study was conceived, co-designed and implemented by a team of three members of the public and one researcher.[24] The primary aims of the study were to estimate the annual and 3-year frequency of patient-reported potentially harmful preventable problems occurring in primary care as described by patients and describe the type of problem. The secondary aims were to identify patient predictors of reporting a problem (eg, age, gender, social class, income, employment status, ethnicity, to describe the primary care service involved), how the problem was discussed (if it was), patient suggestions as to how it might have been prevented and the variation in opinion between the reporting patient, other

---

**Box 1  Brief summary of questionnaire (see the online supplementary appendix 1 for full version). GP, general practitioner; NHS, National Health Service.**

Q1. Did you have confidence and trust in the GP you saw or spoke to at your last appointment?
(benchmarking question)

Q2a. Have you experienced a situation with a primary care service where your health has ACTUALLY been made worse by a problem or error that could have been prevented?

Q2b. And have you experienced a situation with a primary care service where you SUSPECTED your health has been made worse by a problem or error that could have been prevented?

Q2c. And have you experienced a situation with a primary care service where your health could have been made worse had someone not NOTICED a problem or error?

Q2d. And have you experienced a situation with a primary care service where there was a problem or error that could have been prevented but it did not make your health worse?

*If 'yes' to more than one of Q2a-d ask Q2e to identify which happened most recently*

*If 'no' to Q2a-d go to Q11*

Q3. Thinking about the most recent occasion where you experienced a preventable problem or error caused by the primary care service, when did this occur?

Q4. Thinking about the most recent occasion, which primary care service were you using when the problem or error occurred?

Q5. Thinking about the most recent problem or error you experienced, can you briefly describe what it was and how it happened?

Q6. In your opinion, how, if at all, could the problem or error have been avoided?

Q7. Were you able to talk about the problem or error with anybody WORKING IN THE PRIMARY CARE SERVICE?

Q8. You said you were able to discuss the problem or error with somebody working in primary care. Please describe their job or role and their response.

Q9. Which of the following reasons, if any, best describes why you were unable to talk about the problem or error with somebody working in the primary care service?

Q10. In the last 12 months, have any of the following happened to you *while* using primary care, or not? *If yes go to Q4* (see the online supplementary appendix 1 for list of preventable problems).

Q11. Do you, personally, work as a Healthcare Professional in any capacity? For example, a doctor/nurse/therapist/pharmacist/other NHS staff and so on.

---

members of the public and clinicians in their opinion as to the likelihood the patient-described scenario is a potentially harmful preventable problem.

## METHODS
### The population level survey

A survey asking about potentially harmful preventable problems occurring in primary care has been designed and piloted with extensive PPI as described in detail elsewhere.[24] The questions from this survey (box 1, online supplementary appendix 1) were embedded in to the Ipsos MORI Great Britain (GB) Face to Face Omnibus (f2f Omnibus, a weekly survey that is used to

track British attitudes to issues facing the country). It was used to survey a nationally and regionally representative sample of 4000 adults aged ≥15 living in private households in GB between 8 and 21 April 2016 using a random sampling design described elsewhere.[27] Briefly 170–180 geographically representative sampling points were randomly selected and interviewers were required to get the interviews from a small group of streets reflecting that sampling point. (Typically an interviewer would get a completed interview from 1 in every 10–12 addresses.) The sample size was loosely based on the pilot study[24] which had found that 132/638 (21%) of self-selected respondents had perceived a potentially harmful preventable problem (although we anticipated a lower proportion when sampling from the general population). The f2f Omnibus consists of interviews in the participant's home using computer-assisted personal interviewing, participation is completely voluntary and there are no incentives to take part. Respondents are free to refuse to answer any questions. The first question (Q1 box 1) was taken from the English general practitioner (GP) patient survey in order to compare the overall level of confidence and trust in their GP among the survey respondents with the larger sample used in the English GP patient survey.[28] The second question (Q2 box 1) is the main screening question, those responding negatively to Q2 (ie, not experienced a preventable problem) were directed to a more specific question with a list of commonly understood patient safety events (Q10 box 1, online supplementary appendix 1). If this prompted recognition of experiencing a potentially harmful preventable problem they were returned to Q4 (box 1). The intention of using a non-leading screening question was to encourage respondents to express their own perspective on what constitutes potentially harmful preventable problem rather than being directed towards existing definitions.

### Coding of patient-reported scenarios
The nature of the problem described by the patient was coded at face value that is, as the patient described without further interpretation, by one author (SJS) and checked by a second author (JA for dental scenarios, PB for all other scenarios) using a taxonomy developed during the pilot study that also mapped on to a previously published taxonomy for errors in general practice[24 29 30] (Table A, online supplementary appendix 1). The medication-related scenarios were coded to a finer level (Table B, online supplementary appendix 1).

### Likelihood the scenario described a potentially harmful preventable problem
Five GPs, one general dental practitioner and seven members of the public estimated the likelihood that, in their opinion, each patient-described scenario was a potentially harmful preventable problem.[24] The dental scenarios were only rated by the general dental practitioner and members of the public. The raters were given the responses to Q2 and Q4–Q9 (box 1) without

any demographic information and asked to score each scenario on a 5-point scale from 'very likely or certain' to 'definitely not' a potentially harmful preventable problem. The scores were used to categorise the scenarios into two groups according to the public or clinician-estimated likelihoods that they were a potentially harmful preventable problem as below. This is described in detail in table C in the online supplementary appendix 1 and individual coding is shown in the online supplementary appendix 2.

► Group 1: patient-described scenarios with higher threshold as to likelihood of potential harm; median score of 'very likely or certain' or 'probably' or at least one person gave a score of 'very likely or certain'.
► Group 2: patient-described scenarios with lower threshold as to likelihood of potential harm; median score of 'possibly' or at least one person gave a score of 'probably' or higher.
► All other scenarios: median score <3 ('possibly') and 0 scores >3 ('possibly').

The median scores excluded responses where the raters scored 'do not know' or 'insufficient information'. We combined all the patient-described scenarios occurring in the last 3 years with scenarios from the pilot study[24] occurring in the last 12 months. We judged this acceptable since we were using the scenarios to compare the views of the clinicians and members of the public without making any inference to the wider population.

### Statistical analysis
The 95% CIs for the population means were calculated assuming a normal distribution for the sample mean. Simple cross tabulations were used to describe the data and a binary logistic regression model was used to explore whether particular types of patient (eg, according to their demographics or surveyed opinions) were more likely to perceive a potentially harmful preventable problems and what type of scenario was more likely to be ranked as potentially harmful by clinicians and members of the public. Comparisons between demographics and outcomes for the respondents and the UK population were made using a $\chi^2$ test. Inter-rater agreement for the ranking of the patient-described scenarios by clinicians and members of the public was assessed using a two-way random effects model single-measures intraclass correlation coefficient (ICC).[31] All analyses were done using Stata V.14.

### Public and patient involvement
Public and patient involvement (PPI) was central to this co-designed survey and was provided through the Greater Manchester Primary Care Patient Safety Translational Research Centre Research User Group and other PPI networks.[24] The study was conceived, designed, implemented and analysed by a team of three members of the public (AD, CG, JB) and one researcher (SJS). The piloting of the survey was through existing PPI networks.[24] The scoring of the questions as to the likelihood they described a potentially harmful preventable problem was

undertaken by seven members of the public, two of whom had no previous experience in PPI. These findings will be disseminated to all the PPI groups that contributed to the pilot study and the authors will forward these results to their personal contacts who contributed to the questionnaire design.

## RESULTS

Of 3996 members of the public participating in the f2f Omnibus, 3984 (99.7%) agreed to complete the questions relevant to this study and 3975 (99.5%) actually completed all the questions. Survey responders were broadly representative of the GB population but were significantly more likely to have confidence and trust in the GP seen at their last appointment than the English population (Table D, online supplementary appendix 1) although there was no significant difference when the graded responses 'yes definitely' or 'yes to some extent' were combined (91% vs 92%, p($\chi^2$)=0.2).

The progress of the respondents through the analysis is summarised in figures A and B in the online supplementary appendix 1. In total, 300 (7.6%) of respondents reported experiencing a potentially harmful preventable problem during the past 12 months; of these, 193 (4.9%) arose directly from the screening question (Q2 box 1) and 107 (2.7%) were prompted by a list of potentially harmful preventable problems (Q10 box 1, online supplementary appendix 1). Of the 193 unprompted problems (Q2 box 1), 119 (3.0%) patients suspected, or actually believed, that their health had been made worse as a result of the problem, whereas 74 (1.9%) believed that they had either noticed the problem before it had any consequences or it had had no effect on their health. A further 132 potentially harmful preventable problems were reported as occurring within the past 1–3 years (Figure A, online supplementary appendix 1) making a 3-year total of 325 (8.2%) arising only from the screening question (Q2 box 1) as there was no prompt question (Q10 box 1) asking about problems over 12 months ago. The combination of an open-ended question (Q2 box 1) and prompt question (Q10 box 1) prioritised sensitivity over specificity (as intended) given that 21% of the reported problems (79/379) were excluded from being a potentially harmful preventable problem in primary care by the respondent themselves by their response to questions 4 and 6 (ie, not preventable or not in primary care, box 1).

Of the 300 patient-described scenarios occurring within the last 12 months, 93 (31%) were not ranked by any of the six clinicians mostly due to insufficient information (in the clinician's opinion). Of the 207 who were ranked by at least one clinician, 24 (11.6%, Table E, online supplementary appendix 1) were considered to 'at least probably' describe a potentially harmful preventable problem by clinicians (group 1 above). Group 2 (defined above) included 97 (46.9%) scenarios considered to 'at least possibly' describe a potentially harmful preventable problem by clinicians. The members of the public ranked 116 (39%) scenarios occurring in the last 12 months as 'at least probably' a potentially harmful preventable problem (group 1) and this included all 97 scenarios ranked as 'at least possibly' by clinicians (group 2).

The proportion of respondents reporting a potentially harmful preventable problem within the last 12 months by respondent characteristics and unadjusted and adjusted ORs estimated by logistic regression are shown in table 1. Those responding 'no, not at all' to the question about trust and confidence in the GP (Q1 box 1) were around eight times more likely to report a problem compared with those responding 'yes, definitely' (table 1). Women and rural dwellers were significantly more likely to report experiencing a potentially harmful preventable problem even when only including the scenarios judged to be more likely to be potentially harmful by clinicians (table 1). People not in employment due to a disability, self-employed or with one or more children were more likely to report a problem but not when only those scenarios judged to be more likely to be potentially harmful by clinicians were included (table 1).

### Characteristics of the patient-reported scenarios

The types of problem occurring in the last 12 months alongside their clinician rankings are summarised in panel A of figure 1. Generally respondents were equally likely to describe the nature of the problem as related to healthcare delivery, investigation, treatment (mainly medication), communication or lack of clinical knowledge or skills (panel B figure 1). Within the medication problems, the most common scenarios were being prescribed a wrong, contra-indicated or inappropriate drug or the wrong dose or delivery method (panel C figure 1). The respondents did not identify any previously unreported types of problem and the patient-reported scenarios mapped well on to an established taxonomy of errors in primary care (figure 1). However, the prompt question (Q10) particularly increased reports of scenarios related to appointments, referrals and reporting of test results suggesting that the respondents did not consider these to be potentially harmful problems in the first instance (Figure C, online supplementary appendix 1). Table 2 provides information about the patient's response to the potentially harmful preventable problem and the primary care service involved. A substantial minority (30%) of problems occurred outside general practice, particularly the dental surgery, walk in clinic, out of hours care and pharmacy. Around half of the patients had discussed their problem with a primary care professional and usually this was a person who worked in the same organisation as where their problem had occurred (table 2). There were no significant differences between patients who discussed the problem, and those who did not, according to gender (men 49% vs women 51%, p$\chi^2$=0.78), age (38% to 62% in 10-year age bands, p$\chi^2$=0.33), type of service being used (general practice 50% vs other services 50%, p$\chi^2$=0.95),

**Table 1** Prevalence of respondents reporting a potentially harmful preventable problem within the last 12 months and unadjusted and adjusted odds ratios estimated by logistic regression

| Respondent characteristics (total), n=3984 | Reported problem in last 12 months (%), n=300 | Unadjusted OR—all reports | Adjusted[1] OR— all reports | Adjusted* OR after GP review (lower threshold†), n=97 |
|---|---|---|---|---|
| Gender (1 missing) | | | | |
| Male (1950) | 111 (6%) | 1 (ref) | 1 (ref) | 1 (ref) |
| Female (2033) | 189 (9%) | 1.7 (1.3 to 2.2) | 1.7 (1.2 to 2.2) | 2.3 (1.3 to 3.8) |
| Age (years) | | | | |
| 15–24 (533) | 38 (7%) | 1 (ref) | 1 (ref) | 1 (ref) |
| 25–34 (573) | 54 (9%) | 1.4 (0.9 to 2.1) | 0.7 (0.4 to 1.3) | 0.4 (0.2 to 1.2) |
| 35–44 (528) | 30 (6%) | 0.8 (0.5 to 1.3) | 0.4 (0.2 to 0.8) | 0.1 (0.0 to 0.6) |
| 45–54 (629) | 54 (9%) | 1.2 (0.8 to 1.9) | 0.7 (0.4 to 1.4) | 0.5 (0.2 to 1.5) |
| 55–64 (654) | 60 (9%) | 1.3 (0.9 to 2.0) | 0.8 (0.4 to 1.6) | 0.7 (0.2 to 2.0) |
| 65–74 (609) | 41 (7%) | 0.9 (0.6 to 1.5) | 0.5 (0.2 to 1.3) | 0.7 (0.2 to 3.0) |
| ≥75 (458) | 23 (5%) | 0.7 (0.4 to 1.2) | 0.3 (0.1 to 0.9) | 0.3 (0.1 to 1.9) |
| Employment status (3 missing) | | | | |
| Paid job—full or part time (1719) | 119 (7%) | 1 (ref) | 1 (ref) | 1 (ref) |
| Full time student (283) | 14 (5%) | 0.7 (0.4 to 1.2) | 0.4 (0.1 to 1.1) | 0.4 (0.1 to 1.8) |
| Not working—long-term illness/disability (133) | 22 (17%) | 2.7 (1.6 to 4.4) | 2.3 (1.2 to 4.6) | 0.9 (0.3 to 3.1) |
| Not working—other reason (267, includes unemployed) | 24 (9%) | 1.3 (0.8 to 2.1) | 1.3 (0.7 to 2.4) | 0.4 (0.1 to 1.4) |
| Not working—housewife/husband (201) | 19 (9%) | 1.4 (0.8 to 2.3) | 1.0 (0.5 to 2.0) | 0.3 (0.1 to 1.2) |
| Retired (1198) | 80 (7%) | 1.0 (0.7 to 1.3) | 1.4 (0.8 to 2.6) | 0.5 (0.2 to 1.3) |
| Self-employed (180) | 20 (11%) | 1.7 (1.0 to 2.8) | 2.0 (1.1 to 3.5) | 0.5 (0.1 to 2.3) |
| Region of domicile (23 missing) | | | | |
| Greater London (565) | 38 (7%) | 1 (ref) | 1 (ref) | 1 (ref) |
| East Midlands (262) | 9 (3%) | 0.5 (0.2 to 1.0) | 0.6 (0.2 to 1.4) | 0.4 (0.0 to 3.6) |
| East of England (425) | 27 (6%) | 0.9 (0.6 to 1.6) | 0.6 (0.3 to 1.1) | 1.8 (0.5 to 5.8) |
| North (176) | 15 (9%) | 1.3 (0.7 to 2.5) | 0.8 (0.3 to 1.7) | 0.7 (0.1 to 4.3) |
| North-West (490) | 46 (9%) | 1.4 (0.9 to 2.2) | 1.0 (0.6 to 1.9) | 1.4 (0.4 to 4.5) |
| Scotland (372) | 27 (8%) | 1.1 (0.7 to 1.8) | 0.8 (0.4 to 1.6) | 1.8 (0.5 to 6.1) |
| South East (444) | 32 (7%) | 1.1 (0.6 to 1.6) | 1.1 (0.6 to 2.0) | 2.2 (0.7 to 7.0) |
| South West (281) | 33 (12%) | 1.8 (1.1 to 3.0) | 1.0 (0.5 to 2.0) | 1.9 (0.5 to 6.6) |
| Wales (196) | 15 (8%) | 1.1 (0.6 to 2.1) | 0.6 (0.3 to 1.4) | 2.2 (0.5 to 8.5) |
| West Midlands (377) | 19 (5%) | 0.7 (0.4 to 1.3) | 0.6 (0.3 to 1.3) | 1.1 (0.3 to 4.4) |
| Yorks & Humberside (373) | 39 (10%) | 1.6 (1.0 to 2.6) | 1.2 (0.7 to 2.3) | 2.7 (0.8 to 8.4) |
| Ethnicity (18 missing) | | | | |
| White (3591) | 271 (8%) | 1 (ref) | 1 (ref) | 1 (ref) |
| Other ethnicity (475) | 26 (5%) | 0.7 (0.5 to 1.0) | 1.2 (0.7 to 2.2) | 1.1 (0.4 to 3.0) |
| Type of community | | | | |
| Urban, suburban (3051) | 203 (7%) | 1 (ref) | 1 (ref) | 1 (ref) |

Continued

| Table 1 | Continued | | | |
|---|---|---|---|---|
| Respondent characteristics (total), n=3984 | Reported problem in last 12 months (%), n=300 | Unadjusted OR—all reports | Adjusted[1] OR— all reports | Adjusted* OR after GP review (lower threshold†), n=97 |
|---|---|---|---|---|
| Rural (933) | 97 (10%) | 1.6 (1.3 to 2.1) | 1.9 (1.3 to 2.7) | 2.0 (1.1 to 3.5) |
| Parental responsibility | | | | |
| Zero children under 19 (2839) | 192 (7%) | 1 (ref) | 1 (ref) | 1 (ref) |
| Child(ren) aged up to 19 (1145) | 108 (9%) | 1.4 (1.1 to 1.8) | 1.2 (0.8 to 1.7) | 1.5 (0.8 to 2.8) |
| Tenure (31 missing) | | | | |
| Mortgaged (1042) | 84 (8%) | 1 (ref) | 1 (ref) | 1 (ref) |
| Owned outright (1441) | 87 (6%) | 0.7 (0.5 to 1.0) | 0.8 (0.5 to 1.2) | 0.9 (0.4 to 1.8) |
| Rented housing association (301) | 42 (14%) | 1.8 (1.2 to 2.7) | 1.3 (0.7 to 2.2) | 1.1 (0.4 to 2.9) |
| Rented private landlord (719) | 49 (7%) | 0.8 (0.6 to 1.2) | 0.9 (0.6 to 1.5) | 0.9 (0.4 to 2.1) |
| Rented local authority (422) | 31 (7%) | 0.9 (0.6 to 1.4) | 0.6 (0.3 to 1.2) | 1.0 (0.4 to 2.8) |
| Other[28] | 4 (14%) | 1.9 (0.6 to 5.6) | 2.2 (0.6 to 8.2) | –‡ |
| Confidence and trust in GP at last appointment? | | | | |
| Yes definitely (3031) | 144 (5%) | 1 (ref) | – | – |
| Yes, to some extent (611) | 68 (11%) | 2.5 (1.9 to 3.4) | – | – |
| No, not at all (311) | 88 (28%) | 7.9 (5.9 to 10.7) | – | – |
| Do not know/cannot say[31] | 0 (0%) | – | – | – |

*Adjusted for gender, age, employment status, ethnicity, tenure, region of domicile, type of community, parental responsibility, highest level of education achieved, marital status, social grade and household income.
†See Table E in the online supplementary appendix 1.
‡Zero problems in this category.
GP, general practitioner.

working as a healthcare professional (no 56% vs yes 50%, $p\chi^2$=0.44) or describing a problem ranked higher by clinicians (below lower threshold 50% vs above lower threshold 50%, $p\chi^2$=0.98). Those reporting a problem in the first instance at Q2 (box 1) without prompting were somewhat more likely to have discussed the problem (unprompted 53% vs prompted 43%, $p\chi^2$=0.08), whereas ethnic minorities were somewhat less likely to have discussed the problem (white 51% vs other ethnicity 37%, $p\chi^2$=0.09). Patients who discussed their problem were significantly more likely to 'definitely' have trust and confidence in their GP (Q1 box 1; 61% did discuss their problem vs 39% who did not discuss their problem, $p\chi^2$<0.001). The reasons given for not discussing the problem varied but the most common reasons related to feeling uncomfortable about discussing the problem, being too distressed or ill, being unable to find the appropriate person with whom to discuss the problem or the respondent was unconcerned about the problem. The respondent's suggestions as to how the problem might have been prevented are summarised in table 3. The most

frequent suggestions revolved around quicker access to primary care and investigations and a more participatory role. They rarely identified a particular individual as the problem or made specific suggestions for improvement strategies.

### Comparison of the opinions of clinicians and members of the public about the patient-reported scenarios

The total number of patient-described scenarios available for analysis was 564 (432 from the main survey last 3 years and 132 from the pilot survey in last 12 months) but only 406 (72%) patients provided adequate information for at least one clinician to score the scenario on a 5-point scale as to the likelihood that the patient described a potentially harmful preventable problem (Table C, online supplementary appendix 1). The members of the public scored 426 (76%) of the scenarios. The median scores for each patient-described scenario are shown in figure 2. Members of the public were significantly more likely to designate the patient-described scenarios as potentially harmful preventable problems compared

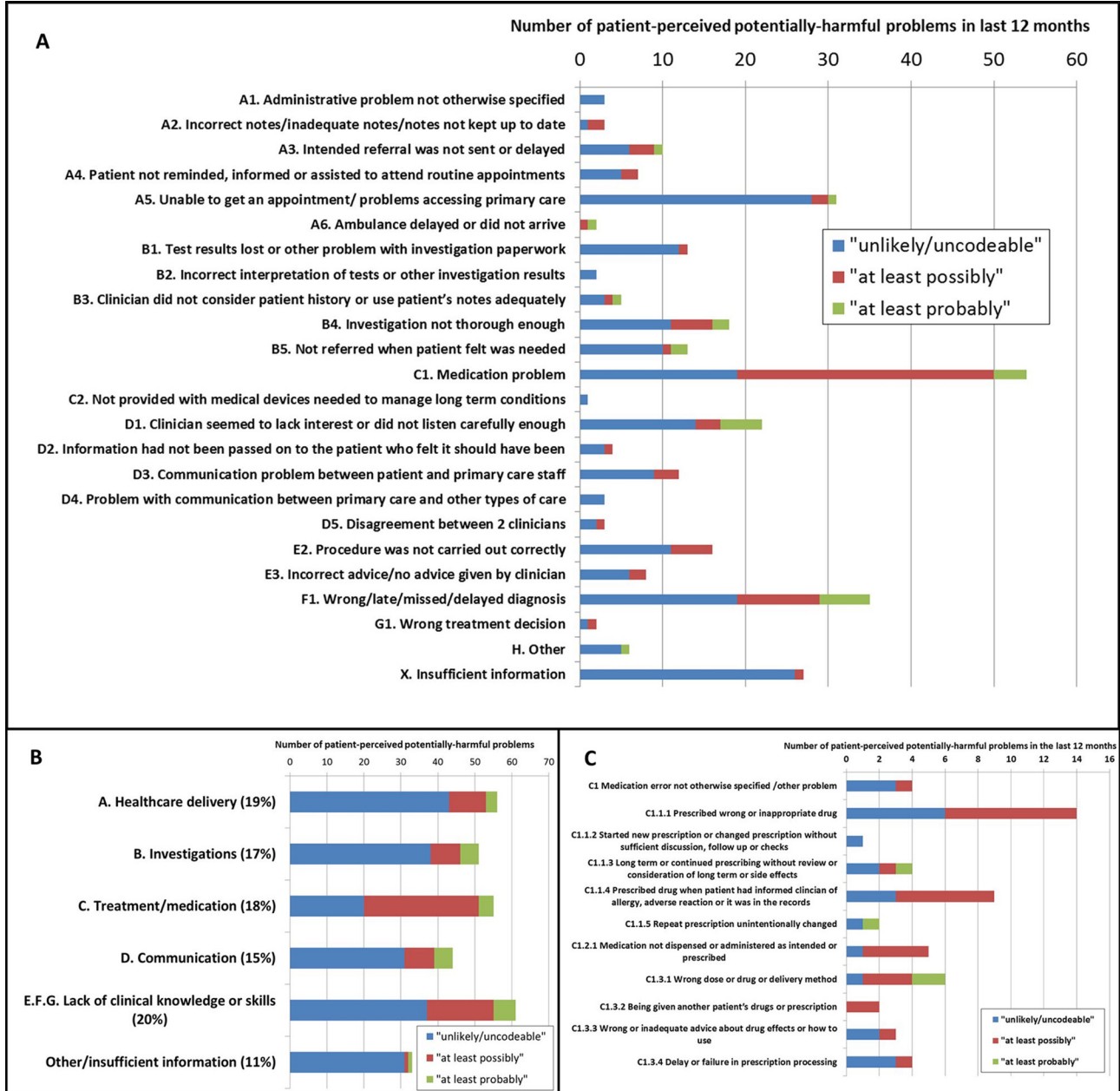

**Figure 1** Numbers of patient-perceived problems occurring in the last 12 months categorised according to the patient's description with clinician ranking as to the likelihood it is a potentially harmful preventable problem (Table E, online supplementary appendix 1).

with clinicians (median clinician score of 2.5, 'unlikely-possibly' compared with members of the public score of 3.5, 'possibly-probably'; Wilcoxon signed-rank test z=16.4, p<0.001). From the clinician perspective, just 8% of the problems occurring during the past 12 months were categorised as 'probably to almost certainly' potentially harmful, whereas for the members of the public, the corresponding proportion was 39% (Table E, online supplementary appendix 1 using the higher threshold). The individual patient-described scenarios scored by clinicians as more likely to be a potentially harmful preventable problems (median score is higher than 'possibly' and scored by at least two clinicians, or one clinician scored 'very likely or certain') and the scenarios

with the greatest disagreement between members of the public and clinicians (median scores differ by ≥3 points on a 5-point scale) are summarised in the online supplementary appendix 2. The single measures ICC for absolute measures was 0.43 (0.38 to 0.49) for the members of the public and 0.23 (0.09 to 0.40) for clinicians, illustrating that members of the public had somewhat better agreement than clinicians. The associations between the characteristics of the patient or problem, and the clinician rankings of the likelihood it is a potentially harmful preventable-problem are shown in table F of the online supplementary appendix 1. Clinicians were more likely to rank scenarios as 'possibly to almost certainly' potentially harmful if they were related to treatment, diagnosis

**Table 2** Details of the patient's response to the potentially harmful preventable problem and the primary care service involved

| Primary care service involved | Problems in last 12 months, n=300 | All problems analysed,* n=564 |
|---|---|---|
| GP surgery | 211 (70%) | 395 (70%) |
| Dental surgery | 27 (9%) | 50 (9%) |
| Walk in clinic | 16 (5%) | 22 (4%) |
| Ambulance/A&E/OOH care | 16 (5%) | 28 (5%) |
| Pharmacy | 10 (3%) | 19 (3%) |
| Community or district nursing | 8 (3%) | 21 (4%) |
| Mental health services | 6 (1%) | 8 (1%) |
| Opticians | 4 (1%) | 5 (1%) |
| Physiotherapy (in primary care) | 2 (1%) | 5 (1%) |
| Missing/nk | 0 (<1%) | 11 (2%) |
| **Did you discuss the problem with primary care staff?** | **Problems in last 12 months, n=300** | **All problems analysed,* n=564** |
| Yes | 145 (48%) | 273 (48%) |
| No | 153 (51%) | 273 (48%) |
| Missing/nk | 2 (1%) | 18 (3%) |
| **Reasons why patients did not discuss the problem with primary care staff** | **Problems in last 12 months, n=153** | **All problems analysed,* n=273** |
| Patient had the opportunity but did not feel comfortable discussing the problem or error | 16 (10%) | 43 (16%) |
| Patient could not find anybody with whom to discuss the problem or error | 37 (24%) | 75 (27%) |
| Patient was not concerned about the problem or error | 25 (16%) | 37 (14%) |
| Patient did not notice the problem or error or trusted the clinician's judgement at the time | 11 (7%) | 25 (9%) |
| Patient was too distressed or ill to discuss the problem or error | 18 (12%) | 30 (11%) |
| Other—problem was resolved in another way by the patient without involving primary care | 10 (7%) | 13 (5%) |
| Other—patient believed primary care staff would not be interested in the problem or would not take it seriously or it would not improve primary care | 7 (5%) | 14 (5%) |
| Other—patient believed that discussing the problem with a primary care staff might have negative implications for their future care | 6 (4%) | 6 (2%) |
| Other—patient did know that they were allowed to express an opinion or how to raise the problem | 5 (3%) | 5 (2%) |
| Other—patient accepts that such problems will arise in primary care or did not want to use primary care resources when primary care staff are very busy | 5 (3%) | 6 (2%) |
| Other—patient intends to discuss with primary care professional at the next opportunity | 4 (3%) | 6 (2%) |
| Do not know/missing | 9 (6%) | 13 (5%) |
| **Profession of discussant** | **Problems in last 12 months, n=145** | **All problems analysed,* n=273** |
| GP/practice nurse | 66 (46%) | 144 (53%) |
| Practice manager/receptionist/administrator | 25 (17%) | 39 (14%) |
| Pharmacist/dispenser | 7 (5%) | 14 (5%) |
| General dental practitioner | 8 (6%) | 18 (7%) |

Continued

**Table 2** Continued

| Primary care service involved | Problems in last 12 months, n=300 | All problems analysed,* n=564 |
|---|---|---|
| Hospital doctor or nurse/A&E or OOH staff/paramedic | 15 (10%) | 18 (7%) |
| Other primary care staff | 14 (10%) | 17 (6%) |
| PALS or NHS direct staff | 1 (1%) | 2 (1%) |
| Unclear, do not know or missing | 9 (6%) | 21 (8%) |
| **Role of discussant in patient's care** | **Problems in last 12 months, n=145** | **All problems analysed,* n=273** |
| Member of staff central to respondent's care | 60 (41%) | 112 (41%) |
| Member of staff in the same team or organisation | 35 (24%) | 84 (31%) |
| Member of staff in a different team or organisation | 31 (21%) | 40 (15%) |
| Role of member of staff is unclear | 8 (6%) | 20 (7%) |
| Missing | 11 (8%) | 17 (%) |

*All problems analysed includes scenarios arising from Ipsos MORI survey in the last 3 years and the pilot survey (24) within the last 12 months.
GP, general practitioner; OOH, out of hours.

or the patient was qualified as a healthcare professional (even though they were blind to this information) but for the members of the public scenarios were related to treatment, investigation, clinical skills, diagnosis or where the patient had reported a problem in the first instance without prompting. Additionally, members of the public were more likely to rank problems reported through the pilot survey as potentially harmful. Potentially harmful preventable problems involving cancer diagnoses or cardiovascular problems were more likely to be considered a potentially harmful preventable problem by both clinicians and members of the public compared with other diagnoses (as specified by the patient).

## DISCUSSION

Our main finding is that 7.6% of respondents in a GB nationally representative survey of 3975 people reported experiencing a potentially harmful preventable problem in primary care during the past 12 months. This is important, not only because patients may be experiencing genuine safety problems, but also because respondents perceiving a potentially harmful preventable problem were found to be eight times less likely to have confidence and trust in their GP (table 1). Furthermore, only around half of these patients perceiving a problem discussed their concern with a primary care professional. The implication is that many patient-perceived problems remain unknown to clinicians—scaling our results up to the GB adult population implies that around 3 million patients (3.8 million; 95% CI 3.3 to 4.2 million) believe that they have experienced a potentially harmful preventable problem during the past 12 months and 1.5 million (1.2–1.8 million) believe or suspect that their health has been made worse as a result. Clearly clinicians need to be aware of these patient-perceived preventable problems where there is the potential for harm, but our findings also suggest that discussing such problems with the

patient may also help to maintain confidence and trust in primary care among those who perceived a problem. (As this is a cross-sectional study, we cannot know whether the patients who discussed their problem did so because they already had a higher level of confidence and trust in their GP or discussing the problem contributed to the higher level of confidence and trust.) An accessible, informal route to actively engage and solicit patient's concerns about primary care may be helpful particularly given that the most common reasons patients gave for not discussing their problems are modifiable, for example, being unable to find the appropriate person or feeling uncomfortable about raising their concern and some were worried about the implications of doing so for their future care. Furthermore, improving communication and patient involvement was one of the most frequently suggested strategies for preventing the potentially harmful preventable problem (alongside quicker access to primary care and investigations). Other work suggested that patients are likely to blame individual clinicians for their perceived problem[7] but we did not particularly find this.

Our finding that around 30% of patient-perceived problems in primary care occurred outside general practice emphasises the need for research in other areas of primary care, for example, 9% of the patient-perceived potentially harmful preventable problems in the last 12 month occurred in dentistry in primary care (corresponding GB estimate 0.34 million; 0.21–0.47 million) yet safety in this area remains largely unexplored.[32 33]

Other studies have found differences between patients in perceiving mistakes or evaluating primary care services according to age, ethnicity, physical health and educational level[34] but we did not find this to be the case. We did find, however, that women, respondents with children, rural dwellers and self-employed people or those not working due to disability were more likely to report a problem (table 1). Some of these groups

**Table 3** Patient suggestions as to how the potentially harmful preventable problem might have been prevented

| How could it be prevented? | Problems in last 12 months, n=300 | All problems analysed,* n=564 |
|---|---|---|
| More resources—total | 100 (33%) | 157 (28%) |
| Quicker access to primary care | 43 (14%) | 62 (11%) |
| More thorough and quicker investigations | 35 (12%) | 59 (10%) |
| Fewer demands on primary care—more staff or fewer patients | 7 (2%) | 12 (2%) |
| More time with clinicians for treatment and diagnosis | 8 (3%) | 12 (2%) |
| Improved access to social care | 3 (1%) | 3 (1%) |
| More follow-up by primary care | 2 (1%) | 3 (1%) |
| Improved continuity of care | 1 (<1%) | 2 (<1%) |
| Access to a second opinion | 1 (<1%) | 2 (<1%) |
| Provision of resources to manage long-term conditions | 0 | 2 (<1%) |
| Improved communication and involvement of patients—total | 53 (18%) | 92 (16%) |
| Listen to the patient and trust their judgement more | 36 (12%) | 68 (12%) |
| Tell patients about their diagnosis, test results, changes in medication or loss of results | 10 (3%) | 15 (3%) |
| Improve communication between staff (within or outside primary care) | 7 (2%) | 9 (2%) |
| Better organisation and administration—total | 27 (9%) | 48 (9%) |
| Follow-up referrals and appointments to ensure they happen, be consistent in sending routine reminders | 12 (4%) | 23 (4%) |
| Log in or process results as soon as received to avoid loss | 5 (2%) | 7 (1%) |
| Keep the notes up to date, well-organised, safe and ensure information is transcribed accurately | 9 (3%) | 15 (3%) |
| Keep a record of the location of equipment | 0 | 1 (<1%) |
| Improve the method of appointment allocation | 0 | 1 (<1%) |
| Fine patients for not attending appointments | 1 (<1%) | 1 (<1%) |
| Improved prescribing systems—total | 21 (7%) | 45 (8%) |
| More when checks on prescribing and dispensing | 19 (6%) | 32 (6%) |
| Check repeat prescriptions carefully, especially for transcribing errors | 2 (1%) | 10 (2%) |
| Use medication reviews and IT clinical decision support systems | 0 | 3 (1%) |
| Better clinical practice—total | 17 (6%) | 47 (8%) |
| Take in to account all the patient's information - their medical history and results and letters | 7 (2%) | 27 (5%) |
| Address the patient's problem in some way—patients can feel their problem is being ignored | 9 (3%) | 18 (3%) |
| Act on advice from other clinicians and test results | 1 (<1%) | 2 (<1%) |
| Staff training—total | 22 (7%) | 53 (9%) |
| More informed and better trained staff | 22 (7%) | 53 (9%) |
| Other responses—total | 60 (20%) | 122 (22%) |
| Do not know/missing | 28 (9%) | 64 (11%) |
| Problem was due to an individual member of staff | 6 (2%) | 11 (2%) |
| Do not make wrong, late, delayed diagnosis | 7 (2%) | 15 (3%) |
| Prescribe right, better, different, more, less medicine | 8 (3%) | 15 (3%) |
| Should have been referred | 6 (2%) | 9 (2%) |
| Better organisation | 3 (1%) | 4 (1%) |
| Patient recognised their own responsibility | 2 (1%) | 2 (<1%) |
| Laboratory procedures were the problem | 0 | 2 (<1%) |

*All problems analysed includes scenarios arising from Ipsos MORI survey in the last 3 years and the pilot survey (24) within the last 12 months.

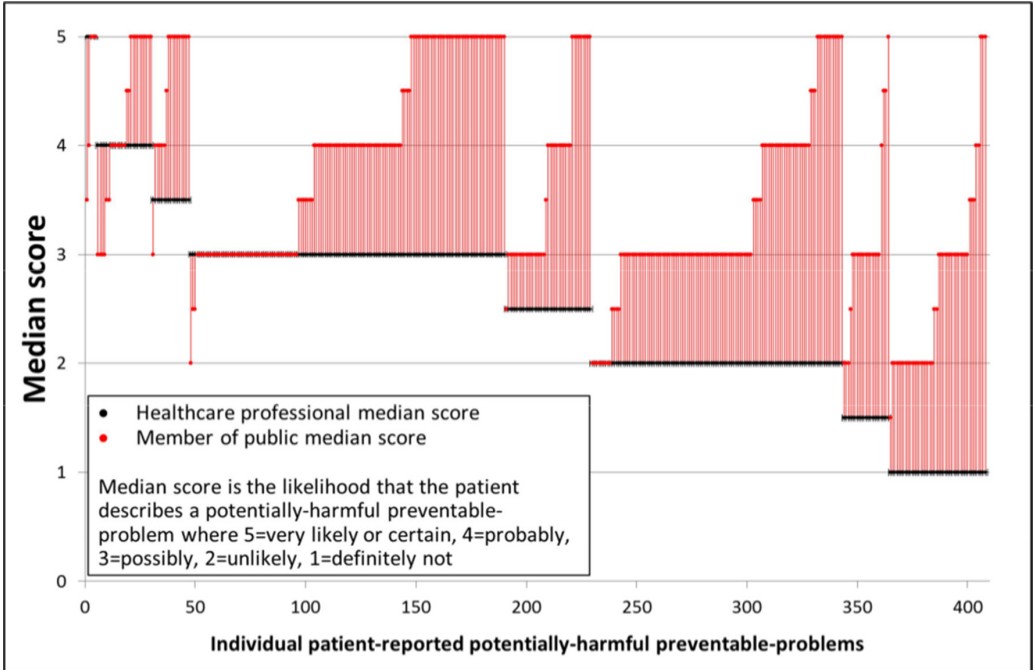

**Figure 2** Median clinician and members of the public estimates of the likelihood that the patient describes a potentially harmful preventable problem occurring in the last 12 months.

might be more frequent users of primary care; in the pilot study, we observed that more frequent users of primary care were more likely to report experiencing a problem.[24] We also observed that respondents identifying with an ethnic minority group were less likely to discuss their problem with a member of primary care staff. Previous work in secondary care suggested that gender, educational level and employment status were associated with a patient's willingness to question healthcare staff.[35] Generally, there were only small differences in demographics between patients in terms of being more or less likely to perceive, or discuss, a problem and it is important to consider each person's problem equally and encourage all groups, including minorities, to share their concerns.

We found that the survey respondents had similar views to clinicians and researchers in what constituted a potentially harmful preventable problem given that the patient-described scenarios fit well in to a taxonomy designed and used by clinicians and researchers.[26 29 30] We did not identify any new types of potentially harmful preventable problems unique to the patients' perspective in primary care. Furthermore, the clinicians and members of the public were consistent in which scenarios they ranked as more likely to be potentially harmful but patients have a much lower threshold for concern than clinicians, for example, just 8% of the 300 patient-reported scenarios were ranked by clinicians as 'at least probably' a potentially harmful preventable problem, whereas for the members of the public it was 39%. While this may not be surprising, it is important in the context of the discussion above. Clinicians may need to address patient-perceived problems that they

do not believe to be harmful if they seek to improve public confidence and trust in primary care.

### Strengths and weaknesses of the study

This large population-level survey allowed for generalisable estimates of the frequency of patient-perceived potentially harmful preventable problems in primary care in GB for the first time and highlights that primary care clinicians tend to judge that the patient-perceived problems are unlikely to be potentially harmful. We have verified that our survey population is similar to the English population in terms of their confidence and trust in their GP as reported in the English GP Patient survey. Previous UK studies[26] have recruited through GP practices whereby patients may be reluctant to disclose problems or answer honestly in case of compromising the patient–clinician relationship; indeed we report here that some patients did not wish to discuss their concern with primary care staff for this, and similar, reasons. Furthermore, we believe that we have comprehensively captured the patient perspective through involving members of the public as research partners from study design through data acquisition to analysis and reporting.[24] We collected data related to problems occurring over the last 3 years and our denominator is patients not consultations. Time is an important tool for a primary care clinician but also problems arise over time, and the time of occurrence cannot always be assigned to a single consultation, especially with errors of omission that are associated with greater harm in primary care.[36] Reporting adverse events at a rate per consultation does not reflect the reality of the patient journey in primary care where the concept of patient

safety as the management of risk over time fits well with the longer time scales.[2] The use of time in this way needs to be communicated to patients given that the most frequently suggested strategy for preventing the problem was quicker access to primary care including investigations (26%, table 3).

The main weakness of the study is the relatively high proportion of scenarios that did not provide adequate information for ranking by clinicians (in their opinion). Arguably this would be improved by using a clinically trained interviewer but this could have biased the scenarios towards the clinician perspective and problems occurring outside of general practice might have gone unnoticed. Furthermore, the cost of employing clinician interviewers would have been prohibitive for such a large-scale survey. Ipsos MORI interviewers are accustomed to asking questions about healthcare; indeed they administer the annual GP patient survey.[28] Perhaps this could have been mitigated by using a more detailed questionnaire but the resources were not available and a longer questionnaire might have reduced the completion rate. A further weakness is that the patients' suggestions regarding prevention tended to be non-specific. Collecting patients' suggestions about preventing harm was a secondary aim of this survey but patients did engage with the question and further work in partnership with clinicians is needed to develop this aspect of the survey further.

### Strengths and weaknesses in relation to other studies

There are few studies undertaken from the patient's perspective at the population level but the annual rates are similar to a Spanish study (7.6% vs 7%).[17] A Health Foundation research scan estimated a 1%–2% adverse event rate per consultation[37] similar to our finding following clinician review (although we do not use consultations as the denominator). A face-to-face interview in family practice waiting rooms in the USA reported that 16% of respondents believed a physician had made a mistake in their care.[38] The types of problem and patient responses to the problem are similar to those that have been described qualitatively[1 21 39 40] but we have taken this further by using a well-defined denominator to quantify the frequency of occurrence and other descriptors of the problem from the patient's perspective.

### Meaning of the study: possible explanations and implications for clinicians and policy-makers

There are potentially a large number of patients in GB who believe they have experienced a potentially harmful preventable problem in primary care but, based on the problems described by patients in this study, primary care clinicians rarely agree that these problems are likely to be potentially harmful. There are already many initiatives in UK primary care aiming to address patient safety but how do we address the patient-perceived problems that clinicians do not

recognise as potentially harmful? Similar differences have been observed in UK secondary care where staff measures of patient safety culture were not correlated with patient measures.[41] These differing views are likely to be multifactorial in nature, for example, perhaps clinicians are considering the problem from a medico-legal perspective or as a matter of allocation of limited resources, for example, disagreement about whether emotional discomfort or wasted time constitutes patient harm?[42] Conversely have the members of the public prioritised sensitivity over specificity or taken a more precautionary approach. Previous qualitative work has observed that, for patients, safety in primary care safety is contingent on the clinician patient relationship where among professionals, the systems approach to patient safety is prevalent.[1] While reconciling the differing perspectives of patient and clinician may not be realisable, our study suggests that providing opportunities for, and encouraging, patients to discuss their concerns informally with a member of the primary care team may help with building trust, clarifying expectations and ensuring understanding. The patient suggestions for preventing their perceived problem seem to be asking for more patient-centred care where healthcare is in partnership and patients are included in decisions.[43] Including patients more actively in healthcare may also help diminish the patient's expectations of certainty that seem to be common despite primary care being inherently uncertain.[44] Future work should focus on strategies to encourage patients and clinicians to work together to ensure that primary care not only is safe but is also perceived to be safe by patients.

### Author affiliations

[1]Centre for Epidemiology, Division of Population Health, Health Services Research and Primary Care, University of Manchester, Manchester, UK
[2]Research User Group (RUG) of the NIHR Greater Manchester Primary Care Patient Safety Translational Research Centre, University of Manchester, Manchester, UK
[3]NIHR Greater Manchester Primary Care Patient Safety Translational Research Centre, Division of Population Health, Health Services Research and Primary Care, University of Manchester, Manchester, UK
[4]Medical Directorate, NHS Greater Glasgow and Clyde, NHS Education for Scotland, Glasgow, UK
[5]St Gabriels Medical Centre, Manchester, UK
[6]Central and South Manchester Specialty Training Programme for General Practice, Health Education England North West (HEENWE) Education and Research Centre, Wythenshawe Hospital, Manchester, UK
[7]Woodlands Dental Practice, Wirral, UK
[8]NHS Education for Scotland, Glasgow, UK
[9]Institute of Health and Wellbeing, University of Glasgow, Glasgow, UK

**Acknowledgements** The authors would like to express their thanks and appreciation for the work done by the Mary Aldred, Gitanjali Holt, Manoj Mistry, Carole Bennett and Lindsey Brown in coding the patient-described scenarios. They also thank the PPI groups who were involved in the piloting of the survey; HelpBeatDiabetes, The Primary Care Research in Manchester Engagement Resource, Associate Research User Group of the Greater Manchester Primary Care Patient Safety Translational Research Centre, The Nowgen Centre, The Citizen Scientist project and North West People in Research Forum. For more information see http://bmjopen.bmj.com/content/bmjopen/8/2/e017786.full.pdf

**Contributors** SJS, AD, JB and CG: conceived and designed the study. SJS, AD, JB, CG, AE, PB, JA, DT, SL, AD, RD and NM: analysed the data. SJS: wrote the

manuscript, and is guarantor. AD, JB, CG, AE, PB, JA, DT, SL, AD, RD, NM and SC: edited the manuscript.

**Funding** This study was funded by the National Institute for Health Research (http://www.nihr.ac.uk) through the Greater Manchester Primary Care Patient Safety Translational Research Centre (NIHR GM PSTRC), grant number gmpstrc-2012-1.

**Disclaimer** The views expressed are those of the author(s) and not necessarily those of the NHS, the NIHR or the Department of Health. The funders had no role in study design, data collection and analysis, decision to publish or preparation of the manuscript.

**Competing interests** None declared.

**Ethics approval** University of Manchester Ethics Committee 2 Approval 15372. Respondents to the Ipsos MORI face to face omnibus are not asked to sign a consent document, the invitation into the house after agreement to take part in the survey is considered to be consent. All respondents were provided with the participant information sheet before completing the survey questions specific to this study which explains that participation is entirely voluntary and the participant may choose to stop answering the questions at any time.

**Provenance and peer review** Not commissioned; externally peer reviewed.

**Data sharing statement** Raw data (coded only) are available from sjstocks@btinternet.com

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
