## [Reviewer comments · BMJ Open]

This paper was submitted to a another journal from BMJ but declined for publication following peer review. The authors addressed the reviewers' comments and submitted the revised paper to BMJ Open. The paper was subsequently accepted for publication at BMJ Open.

ARTICLE DETAILS

TITLE (PROVISIONAL)	The frequency and nature of potentially-harmful preventable-problems in primary care from the patient's perspective with clinician review – a population level survey in Great Britain
AUTHORS	Stocks, Susan; Donnelly, Ailsa; Esmail, Aneez; Beresford, Joanne; Luty, Sarah; Deacon, Richard; Danczak, Avril; Mann, Nicola; Townsend, David; Ashley, James; Gamble, Carolyn; Bowie, Paul; Campbell, Stephen

VERSION 1 – REVIEW

REVIEWER	Watt, Ian The University of York, Department of Health Sciences
REVIEW RETURNED	19-Jun-2017

GENERAL COMMENTS	This is a clearly written article which reports the results of a simple but well undertaken population level survey. It presents a frequency of patient perceived potentially harmful problems occurring in primary care. There is little existing research in this area and the paper represents a welcome addition to the literature. The study is a large population level survey with an excellent response rate. The decision to collect data relating to problems occurring over a 3 year period and use patients as a denominator rather than consultations is to be welcomed. The finding that 30% of patient perceived problems occurred outside general practice is important as it draws attention to the need to undertake research on safety in other areas of primary care such as dentistry. These other areas have been relatively unaddressed to date by patient safety researchers. If I had any minor criticism of the paper is that the text detailing some of the results is a little difficult to follow at times and I wonder whether a brief summary at the end of the results section would help. Otherwise I find this a clear description of a well undertaken study and would support its publication
---

REVIEWER	Heyhoe, Jane Bradord Institute for Health Research, Quality and Safety
REVIEW RETURNED	17-Aug-2017

GENERAL COMMENTS	This paper reports interesting research which uses findings from a GB population level survey to estimate the frequency, and describe the nature, of potentially harmful, preventable problems in primary care from the patient perspective.
--

	The large sample, detailed methods and analyses provides convincing evidence of the frequency and scale of patient-perceived mistakes and problems in primary care. The authors' aim of capturing the patient perspective is supported by strong PPI in all aspects of the study. The finding that patients were able to describe problems in a similar way to the categories used in the taxonomy for errors in general practice and the suggestion that "patients and clinicians have a shared understanding of what constitutes a potentially-harmful preventable-problem" is particularly interesting and has implications for future work exploring the role of patients in improving care and safety in a primary care setting. With this in mind, I think the paper would benefit from the authors incorporating a more structured consideration of findings and the implications for safety and the reduction of mistakes and problems in primary care. As a reader I found the paper a bit disjointed and lacking coherence in its current form. At times I found myself asking what the real messages were despite some novel findings and learning points. For example, while perceived frequency and nature of problems are clearly illustrated, the authors also state "our study goes further than describing and counting the frequency of occurrence of potentially-harmful preventable-problems by providing information about how patients dealt with the problem and how it might be prevented." While the authors do discuss their findings on the link between patient trust and confidence and patients discussing problems with a primary care professional, it might also be interesting to incorporate further discussion on patients' ideas around prevention and how this knowledge may be useful for moving forward in terms of "healthcare as a partnership" and improving safety. Some additional reflection on how this study highlights what knowledge is missing and suggestions of ways to fill these gaps would also be helpful.
--	---

VERSION 1 – AUTHOR RESPONSE

Reviewer: 1

Comments to the Author

Reviewer 1: This is a clearly written article which reports the results of a simple but well undertaken population level survey. It presents a frequency of patient perceived potentially harmful problems occurring in primary care. There is little existing research in this area and the paper represents a welcome addition to the literature. The study is a large population level survey with an excellent response rate. The decision to collect data relating to problems occurring over a 3 year period and use patients as a denominator rather than consultations is to be welcomed. The finding that 30% of patient perceived problems occurred outside general practice is important as it draws attention to the need to undertake research on safety in other areas of primary care such as dentistry. These other areas have been relatively unaddressed to date by patient safety researchers.

Authors' response: Thank you for the positive comments

Reviewer 1: If I had any minor criticism of the paper is that the text detailing some of the results is a little difficult to follow at times and I wonder whether a brief summary at the end of the results section would help.

Authors' response: We welcome this suggestion and agree that the large quantity of information included in the results section makes it difficult to follow. We have added a paragraph from p7 line 4 to p8 line 29 at the end of the results section to summarise the results which is specifically linked to the aims as suggested by the Associate Editor. However this has added around 450 words to the paper and could be removed at the discretion of the Editors. We have also made other changes to the results section which we believe makes it easier to follow.

Reviewer 1: Otherwise I find this a clear description of a well undertaken study and would support its publication

Authors' response: Thank you

Reviewer: 2

Comments to the Author

Reviewer 2: This paper reports interesting research which uses findings from a GB population level survey to estimate the frequency, and describe the nature, of potentially harmful, preventable problems in primary care from the patient perspective.

The large sample, detailed methods and analyses provides convincing evidence of the frequency and scale of patient-perceived mistakes and problems in primary care. The authors' aim of capturing the patient perspective is supported by strong PPI in all aspects of the study. The finding that patients were able to describe problems in a similar way to the categories used in the taxonomy for errors in general practice and the suggestion that "patients and clinicians have a shared understanding of what constitutes a potentially-harmful preventable-problem" is particularly interesting and has implications for future work exploring the role of patients in improving care and safety in a primary care setting.

Authors' response: Thank you we agree that this is an important observation

Reviewer 2: With this in mind, I think the paper would benefit from the authors incorporating a more structured consideration of findings and the implications for safety and the reduction of mistakes and problems in primary care. As a reader I found the paper a bit disjointed and lacking coherence in its current form. At times I found myself asking what the real messages were despite some novel findings and learning points.

Authors' response: Thank you, we agree that we could have done a much better job of guiding the reader through the paper and interpreting the results in the discussion. We have taken these comments on board and substantially reorganised the discussion to emphasise our main findings and what can be learnt from this study.

For example, while perceived frequency and nature of problems are clearly illustrated, the authors also state "our study goes further than describing and counting the frequency of occurrence of potentially-harmful preventable-problems by providing information about how patients dealt with the problem and how it might be prevented." While the authors do discuss their findings on the link between patient trust and confidence and patients discussing problems with a primary care professional, it might also be interesting to incorporate further discussion on patients' ideas around prevention and how this knowledge may be useful for moving forward in terms of "healthcare as a partnership" and improving safety. Some additional reflection on how this study highlights what knowledge is missing and suggestions of ways to fill these gaps would also be helpful.

Authors' response: Our finding about the relationship between perceiving a problem and lowered confidence and trust in their GP which is mitigated by discussing the problem with a primary care

professional is important. We agree that we should have included more discussion of patient's ideas around prevention and we have done so as far as possible although the information provided tended to be fairly non-specific. Generally few clinicians would disagree with the common sense advice made by patients although it is reassuring that patients tried to be constructive in their suggestions, for example they rarely identified a particular individual or clinical practice as requiring improvement. This might be because this was the final question but we suspect it was also because many patients felt that the way to prevent the problem was inherent in their description e.g. if the clinician had not done something then he/she should have done it and so on. So we think that this question needs to be asked in a different way, perhaps in a more general context rather than this particular scenario, and needs some consideration. We have added to the discussion as pasted below on p10 lines 1-3 to reflect this.

P10 lines 1-3 "Collecting patients' suggestions about preventing harm was part of aim ii of this survey but the suggestions tended to be non-specific. However patients engaged with the question and further work in partnership with clinicians is needed to develop this aspect of the survey."

Associate Editor Comments to the Authors

Associate Editor: Two experts in the field have now reviewed this manuscript. Both agree that this is a novel and well executed study and both recognise that it addresses a gap in the current evidence base. I agree, wholeheartedly with the reviewers that this work makes an important contribution to the field and I encourage you to make the minor revisions suggested by the reviewers and myself.

Authors' response: Thank you for the encouraging comments. We have done our best to respond to the suggestions made by yourself, the Editor-in -chief and the reviewers and believe that the paper is much more coherent and easier to follow than the previous version.

Abstract

Associate Editor: 'in' missing from the first sentence after 'occurring'

Associate Editor: unnecessary 'in' A substantial minority (30%) of problems occurred 'in' outside primary care....

Authors' response: Thank you for pointing this out, it has been corrected.

The conclusion is a speculative comment rather than something that is based on the findings of your research

Authors' response: We have changed the conclusion in light of your comment and the Editor-in-chief's comments. We do believe that our work supports the original statement but agree it should not be the main conclusion. Our new conclusion is

"Reconciling patient and clinician's views about preventable harm in primary care is important in maintaining patient trust and confidence."

Methods

Associate Editor: In the methods section the authors report that:

Five GPs, one dentist and 7 members of the public estimated the likelihood that, in their opinion, each patient-described scenario was a potentially-harmful preventable-problem.

However, Appendix 4 shows the ratings for each scenario and there appear to be between 1 and 5 GP ratings for each scenario. The authors should articulate this discrepancy and describe how they dealt with those reviews where only one or two GPs provided a rating.

Authors' response: Thank you for raising this. All 5 GPs rated all the scenarios except the dental ones which were rated by one general dental practitioner. The members of the public rated all scenarios. If there is no bar in the plot in online Appendix 4 it means that the GP had rated it as either "insufficient information" or "don't know". Often they explained their reasoning for this judgement but adding this to the analyses in this paper would add still further complexity. It could be the topic of another paper in future. In terms of the median scores for each scenario we only included scores from 1 to 5 i.e. don't know or insufficient information was treated as missing. In order that this did not bias the higher and lower threshold groups by including median scores based on just one score they were defined in a way where one higher score qualified for inclusion as well as the median score (as described in Table B, online Appendix 3). The methods have been altered by adding the text below on px, lines x to x.

P4 lines 39-40 "The dental scenarios were only rated by the dentist and members of the public."

P5 lines 5-6 "The median scores excluded responses where the raters scored "don't know" or "insufficient information"."

Associate Editor: In response to the concerns of reviewer 2 that there are a number of aims of the study and the key messages might get lost a little in this complexity, this could be quite simply addressed by structuring both the Method and Results section according to the aims.

Authors' response: Thank you for this excellent suggestion. The methods section has been labelled with the relevant aim in each section. The results section now starts with a list of which figures and table correspond to each aim and the summary of the results added at the suggestion of reviewer 2 is labelled with the aims. It wasn't really possible to label the entire body of the results section by aim because the same results were sometimes relevant to more than one aim.

Discussion

Associate Editor: I'm not sure that the discussion picks up on all three aims of the study as there is little or no reference to the predictors of patient reported problems. Perhaps a couple of sentences addressing this aim would suffice.

Authors' response: The predictors of patient-reported problems are discussed in a paragraph under the heading "Strengths and weaknesses in relation to other studies" p10, lines 26 to 39. Perhaps the Editor is referring to the predictors of whether a problem will be ranked as more likely to be potentially harmful as this was not discussed. In light of the Editor-in-chief's comments suggesting that we focus more on the differences between clinicians and members of the public we have added the predictors for members of the public to Table C in online Appendix 3 and this has been reported in the results and discussion as shown below

Results P7, lines 32 to 39: "Clinicians ranked a higher likelihood for a potentially-harmful preventable-problem if the problem was related to treatment, communication or diagnosis or the patient was a qualified healthcare professional. Members of the public ranked scenarios more highly if the problem was related to treatment, investigation or diagnosis or the patient believed that their health had been affected. The diagnoses (as specified by the patient) considered more likely to be a potentially-harmful preventable-problem by both clinicians and members of the public were cancer and cardiovascular problems but only the public were likely to consider problems related to mental health, pregnancy and infections as potentially harmful."

Discussion p10 line 45 to p11 line5: "Patient trust and confidence in primary care could be improved by addressing all patient-perceived potentially harmful problems, not only those clinicians believe to be potentially-harmful. For example members of the public tended to rank problems related to mental health, pregnancy and infection as potentially harmful more frequently than clinicians possibly suggesting a need to reassure patients with these diagnoses. Furthermore clinicians were more likely to rank scenarios described by healthcare professionals as potentially harmful even though they were blind to this information. Perhaps healthcare professionals are better at articulating their problems in a way that resonates with clinicians but not members of the public. "

Editor-in-Chief

Comments to the Author

Editor-in-Chief: I have to apologize for the delay in sending this decision to you. We had trouble finding reviewers for your manuscript. This does not in any way reflect on the quality of your work. This happen sometimes and sadly it says more about the willingness of colleagues to support the peer review process than it does about the quality of the manuscript. The summer months, with absences among editors and potential reviewers, compound this year round problem.

Authors' response: We understand and appreciate your efforts to find reviewers.

Editor-in-Chief: On top of that, there was some disagreement among the editors about your paper. the associate editor handling it generally likes it and thinks you need to make some minor revisions (outlined below).

While I hope the outcome of the revision is successful, I have a concern about the coherence of the message of the paper -- one of the reviewers touched upon this problem as well.

Reviewer 2 states at one point:

"As a reader, I found the paper a bit disjointed and lacking coherence in its current form. At times, I found myself asking what the real messages were despite some novel findings and learning points...."

Authors' response: We have reworked the paper to address this comment by reviewer 2 and believe that it is much more coherent following a substantial revision.

Editor-in-Chief: I shared this concern. It's not clear what the message is. In the end, you make it seem as if what you are doing is characterising the epidemiology of harm from the patient's perspective.

Authors' response: We are describing and quantifying patient's perceptions of potentially harmful problems entirely from their perspective then asking for clinicians views. Our message is that patients believe they have experienced a potentially-harmful problem quite frequently whereas clinicians do not agree that it is a potentially-harmful problem for most of patient-reported scenarios. Our main point is that we need to reconcile these differences in order to improve patient confidence and trust in primary care. We apologise if this was not clear in the submitted version and we have substantially re-written the paper to clarify our aims and findings.

Editor-in-Chief: But, how does one reconcile this result with the fact that clinicians didn't agree with most of the harms?

Authors' response: Absolutely yes, this is the natural response to this study and we intend to (and hope this will encourage others to do likewise) work towards reconciling this large difference in patient and clinician opinion about the potential for harm in primary care. We believe that unless we address these differences in opinion we will not be able to improve patients' trust and confidence in primary care. Recognising this large discrepancy in opinions in this study is the first step towards resolving it. We do not suggest that it will be an easy task but it needs to happen if we are to improve patient trust

and confidence in primary care. The next stage in this work is to find out more about how we can help clinicians and patients to understand each other's point of view.

However we must also point out that there was a good agreement between clinicians and patients regarding the nature of a potentially harmful problem. Prior to this work we anticipated that patients might suggest some types of problem that were unique to the patient perspective but this was not the case. Furthermore patients and clinicians rank the same problems as more likely to be potentially-harmful i.e. they are consistent in this respect but the public consistently rank the problems as more likely than the clinicians.

Editor-in-Chief: Let me be clear, I am a clinician and am sensitive to the possibility that clinicians often devalue the patient perspective. I like to think I do not make that mistake – as a clinician or as an editor.

Author's response: We are pleased to hear this, it is not a question of whose view is more important – that will cause an adversarial type of patient-clinician relationship which is not our aim at all – it is question of pointing out the magnitude of the difference in opinion and seeking to find the underlying reasons for this difference in order that we might address it.

Editor-in-Chief: Certainly, we have published many papers on the patient's perspective, including research on patient-centered interventions and even memoirs written by patients about the care they received (e.g., see this editorial on two companion pieces by patients <https://www.ncbi.nlm.nih.gov/pubmed/?term=21964611> and this paper on a registry of patient accounts of serious harms they experienced: .

Southwick FS, Cranley NM, Hallisy JA. A patient-initiated voluntary online survey of adverse medical events: the perspective of 696 injured patients and families. *BMJ Qual Saf.* 2015 Oct;24(10):620-9.)

Author's response: we agree with all the Editor says above and this work is all very important. However our work is very different to these patient-described experiences, not only because they occurred in a secondary care setting whereas our work is in primary care, but because our study is quantitative and generalisable to the GB population. We aimed to quantify the scale of patient-perceived problems and the differences in opinion between patients and clinicians, rather than the more detailed description of a single patient's experience.

We have now cited the Southwick paper as ref 10 in our introduction as an example of a patient-driven study in secondary care. This patient-driven approach has some similarities to the approach we took when developing the survey (ref 26) but we have taken our work further by actively collecting data from a representative sample of the population (rather than the passive data collection as in our pilot work and the Southwick paper) so that we can make generalisable estimates. Again we point out that although the questionnaire used in the Southwick paper (<https://www.surveymonkey.com/r/?sm=p7JEPTM4TYa%2bxOAO1GILMQ%3d%3d>) asks about the type of healthcare setting it does not report specifically on the events occurring in primary care.

Editor-in-Chief: Plus, the patient safety field owes its origin to studies conducted and championed by clinicians – studies showing high rates of harm as judged by clinicians reviewing medical records or capturing adverse events in other ways.

Author's response: yes we agree the vast majority of studies are “championed by clinicians” and we hypothesised that patients might take a completely different approach to measuring harm that could identify types of harm unique to the patients' perspective. As it turns out they did take a different approach but did not identify any type of problem of which clinicians were unaware although they perceived them to be more concerning than clinicians.

Editor-in-Chief: There is also an existing literature showing that there is an overlap or correlation between cases that clinicians identify as having problematic elements of care and about which patients themselves also expressed concerns. E.g., many papers by Saul Weingart and colleagues in Boston in the early to mid 2000s (e.g., Weingart SN et al. RS What can hospitalized patients tell us about adverse events? Learning from patient-reported incidents. *J Gen Intern Med.* 2005 Sep;20(9):830-6.)

Authors' response: Yes this is all important work but it is undertaken in the secondary care setting and led by clinician investigators. Our aim was to look at potential harm from the patient perspective in primary care which we feel is much less frequently the topic of research.

Editor-in-Chief: One of these articles highlights the degree to which conventional medical record review misses many adverse events identified by interviews with patients. But, clinician investigators are still the ones judging these adverse events identified by patient interviews. [Weissman JS, Schneider EC, Weingart SN, et al. Comparing patient-reported hospital adverse events with medical record review: do patients know something that hospitals do not? *Ann Intern Med.* 2008 Jul 15;149(2):100-8.]

Authors' response: Again very interesting work that addresses the discrepancy between patient and clinician views but it occurs in the secondary care setting. In the introduction we state "involving patients in identifying errors and reducing harm is common in secondary care.(3)". Reference 3 is a review and it is true that we could have cited papers not included in the review or published after the review (2013) but as we assert that the patient perspective is frequently considered in secondary care we did not think this was necessary. We have added these citations above as refs 8-9 in addition to reference 3 to make sure we acknowledge that the patient's perspective has been considered previously.

Editor-in-Chief: My point is not that the clinicians' perspective is the only one to consider. But, it's hard to ignore the discrepancy between clinicians' and patients' perspectives on the frequency of harm in your study.

Authors' response: We absolutely do not seek to ignore this, it is our main finding alongside the frequency with which patients perceive potentially harmful problems. We apologise if this did not come through clearly enough and have tried to make sure it is very clear now.

Editor-in-Chief: This discrepancy undermines the epidemiologic findings, which are what you seem to highlight at the main message currently.

Authors' response: The point of our paper is that many patients perceive they have experienced a potentially harmful problem that is not recognised as harmful by a clinician. We have altered the description of our main findings to emphasise this point. We believe that this is extremely important because we show it impacts on patient trust and confidence in clinicians which is an important component in delivering safe care. Furthermore confidence in primary care is an aspiration of the UK NHS as evidenced by the high profile given to the annual GP patient survey.

Editor-in-Chief: It seems like an easier hook for your paper and more coherent package for your work to embrace the aspect of your study having vignettes considered by patients and by clinicians. This is more interesting and novel than the results and conclusions at present, which come across as showing that there is such-and-such level of harm in primary and we need to do this-or-that about it.

Authors' response: We have done exactly this, we have given equal weight to this aspect of the study as to the estimates of patient-perceived harm. We believe both are novel and important results. Our

conclusions are not as you suggest, we always refer to patient perceptions and that we need to address patient perceptions to bring them more in to line with clinicians' views.

Editor-in-Chief: We already think we need to do something about patient safety. No one is questioning that.

Authors' response: we are not suggesting that there is no interest in this topic but the point is that many patients think they are not receiving safe care when it appears, at least from the clinicians' perspective, that the care is safe. So surely it makes sense to recognise and address this perception, it must be important not only that care is safe but that patients perceive it to be safe. Furthermore we would suggest that loss of trust and confidence in primary care can be a safety problem in itself.

Editor-in-Chief: And, as a journal, we have promoted studies of harm in primary care settings, including the systematic review by Panesar (which you cite as your ref 11), but also many others (which you do not cite):

Authors' response: We already had 38 citations of which 6 were reviews or evidence scans that included some of the literature listed below and one book. We were not aiming to comprehensively list all papers that estimate of the frequency of errors or safety problems in primary care but to show that the published estimates are pretty much all from the clinician's perspective and give an idea of their frequency of occurrence from that perspective. We think that the review by Panesar adequately shows this but we are happy to add further citations to emphasise this still further. However we could not list all studies of harm in primary care but we have added all of those below that seem to be relevant to our work.

Citation suggested by Editor-in-Chief: de Wet C et al. Implementation of the trigger review method in Scottish general practices: patient safety outcomes and potential for quality improvement. *BMJ Qual Saf.* 2017 Apr;26(4):335-342.

Authors' response: The aim of this paper was to report the implementation of a trigger review method in primary care so while it does quote the prevalence of specific safety incidents these are not easily comparable to the other estimates. However the characteristics of the PSIs are similar to those described in Figure 1 in our paper providing further evidence that patients identify similar types of incident as clinicians. (We were certainly aware of this paper because 2 of the authors on the paper under review were also authors on this paper by de Wet et al.).

Citation suggested by Editor-in-Chief: Singh H, Meyer AN, Thomas EJ. The frequency of diagnostic errors in outpatient care: estimations from three large observational studies involving US adult populations. *BMJ Qual Saf.* 2014 Sep;23(9):727-31

Citation suggested by Editor-in-Chief: Singh H, Schiff GD, Graber ML, Onakpoya I, Thompson MJ. The global burden of diagnostic errors in primary care. *BMJ Qual Saf.* 2017 Jun;26(6):484-494

Authors' response: These papers are very interesting. We did not include them because they focus on a particular type of error, diagnostic error. As we mention above medical error is a very wide field so if we start to discuss different types of error in the introduction it will become very long. If there had been any estimates of diagnostic error rates in the UK or GB (from any perspective) that we could have compared with our data in the discussion we would have done so but the authors actually only quote a 5% diagnostic error rate for the USA and state that "... it is unclear whether this rate would be similar in primary care in other countries, where data are generally lacking." We would be happy to cite this work if the editor still thinks it is appropriate.

Citation suggested by Editor-in-Chief: Litchfield I, et al. Routine failures in the process for blood testing and the communication of results to patients in primary care in the UK: a qualitative exploration of patient and provider perspectives. *BMJ Qual Saf.* 2015 Nov;24(11):681-90.

Authors' response: we have cited this paper as reference 38 alongside references 1, 21, 37 in supporting our statement that the patient suggestions for preventing the problem agree well with those reported previously in qualitative work.

"The types of problem and patient responses to the problem are similar to those that have been described qualitatively (1, 21, 37,38) but we have taken this further by using a well-defined denominator to quantify the frequency of occurrence and other descriptors of the problem from the patient's perspective."

Citation suggested by Editor-in-Chief: Hernan AL, Giles SJ, O'Hara JK, Fuller J, Johnson JK, Dunbar JA. Developing a primary care patient measure of safety (PC PMOS): a modified Delphi process and face validity testing. *BMJ Qual Saf.* 2016 Apr;25(4):273-80.

Authors' response: we did not feel it appropriate to include this work in our introduction because it does not have similar aims to our paper. Furthermore face validity in an Australian population might not be transferable to a UK population. However we have referenced the paper below which cites this paper and is based on the method described in the above paper.

Citation suggested by Editor-in-Chief: Hernan AL, Giles SJ, Fuller J, Johnson JK, Walker C, Dunbar JA. Patient and carer identified factors which contribute to safety incidents in primary care: a qualitative study. *BMJ Qual Saf.* 2015 Sep;24(9):583-93.

Authors' response: we have cited this paper as reference 37 alongside references 1, 21, 38 in supporting our statement that the patient suggestions for preventing the problem agree well with those reported previously as quoted above.

However we feel our findings are stronger because they are based on a survey of almost 4000 individuals thereby offering a sense of how frequently each factor was considered important in preventing a problem whereas this paper bases its conclusions on 4 focus groups comprising 26 people in total.

Citation suggested by Editor-in-Chief: Wallis K, Dovey S. No-fault compensation for treatment injury in New Zealand: identifying threats to patient safety in primary care. *BMJ Qual Saf.* 2011 Jul;20(7):587-91.

This is included in the evidence scan alongside similar literature in reference 33 so we did not cite it separately.

Citation suggested by Editor-in-Chief: Zwart DL, Steerneman AH, van Rensen EL, Kalkman CJ, Verheij TJ. Feasibility of centre-based incident reporting in primary healthcare: the SPIEGEL study. *BMJ Qual Saf.* 2011 Feb;20(2):121-7.

This is included in the evidence scan alongside similar literature in reference 33 so we did not cite it separately.

Editor-in-Chief: I am not being defensive as the editor of a journal which has published a number of relevant papers which you do not cite

Authors' response: we are pleased to hear that

Editor-in-Chief: (though it's never a great sign when authors do not seem aware of work directly relevant to their research).

Author's response: we believe some of this is indirectly relevant for the reasons we describe above and have added the papers where they are relevant. As the authors of some of these papers are our colleagues and co-authors or even authors of these papers we were aware of them, some of them were omitted as a conscious decision but we apologise where there were oversights.

Editor-in-Chief: My point is more along the following lines. There is much more literature than you acknowledge on the frequency of harm experienced by outpatients.

Authors' response: While we agree there is quite a lot of work published from the clinicians' perspective and from the patient's perspective in secondary care, we still believe that there is a shortage of quantitative literature on safety from the patients' perspective in primary care. The work that has been done is almost all qualitative. Here we provide evidence that some, although not all, of this qualitative information is generalisable to the wider population. Furthermore this work is unusual in its high level of patient involvement, the majority of studies listed above include patients as participants whereas patients were directly involved in designing this study. Co-designed studies such as this are rare and quantitative co-designed studies are even more unusual. However we do accept that we probably overstated the lack of literature and have placed much less emphasis on this in the revised introduction.

Editor-in-Chief: So, you should decide why that aspect of your study – the aspect of your study in which you use patients' assessment of harm as the basis for estimating the epidemiology of safety problems – is the one worth highlighting.

Authors' response: We are not estimating the frequency of safety problems, we are estimating how frequently patients perceive a safety problem. If the views of 6 clinicians (albeit heterogeneous) is taken as the gold standard then we have shown the patient's perception is very different to the reality which surely needs to be addressed for a myriad of reasons. We agree that further work is needed to really understand why there is such a difference in opinions between patients and clinicians, and how to address it, but we have added some insight in to this and it will form the basis for further work.

Editor-in-Chief: Again, the above studies mostly use clinicians or clinician investigators working with patient reported information to provide estimates that are at least as compelling as yours. For instance, one study showed that 10% of the US population will experience a serious diagnostic error [Singh H, Meyer AN, Thomas EJ. The frequency of diagnostic errors in outpatient care: estimations from three large observational studies involving US adult populations. *BMJ Qual Saf.* 2014 Sep;23(9):727-31.]

Editor-in-Chief: The Panesar et al paper, which you refer to as one of just a few existing studies, was in fact a synthesis of 9 systematic reviews and 100 primary studies. So, again, there is no shortage of estimates. And, what this synthesis reported was that studies reported between <1 and 24 patient safety incidents per 100 consultations. The authors provided a median from population-based record review studies of 2-3 incidents for every 100 consultations/records reviewed, and estimated that around 4% of these incidents may be associated with severe harm, defined as significantly impacting on a patient's well-being, including long-term physical or psychological issues or death (range <1% to 44% of incidents).

Author's response: we do not believe that the sentence below implies that ref 14 is a single study and certainly did not intend to do so.

P3 lines 18-19 "Estimates of the incidence of patient safety problems in primary care range from less than 1 to 24 per 100 consultations or record review.(13-15)"

The majority of the primary studies were not of sufficient quality to be included in this review - "Eighty-eight per cent of the systematic reviews (8/9) and 12% of the primary studies (12/100) were judged to be of high quality." I suppose it depends on your perspective – we wouldn't say that a 100 primary studies with only 12 of sufficient quality to contribute to an estimate resulting in very wide confidence intervals is a large body of literature given estimates of over 400,000 publications per year in the field of medicine.

Editor-in-Chief: Unless you make a case for why your study is special, it is just another estimate in this wide range.

Authors' response: It is an estimate based on a representative sample of the population, this has not been done before. We actually have a defined denominator which means that these estimates are generalizable and we are able to quantitatively consider the predictors of the estimates. As we keep pointing out we are not estimating the frequency of safety problems but the frequency of patient-perceived safety problems.

Also this study is one of the first to report on the views of the patient entirely from the patients' perspective rather than within a framework designed by clinicians. In research and clinical practice the clinicians' perspective is undoubtedly the most influential but it is important that the patient's voice is heard entirely from their perspective at least occasionally. Prior to this work we did not know whether patients might identify unrecognised type of problem or prioritise problems very differently to clinicians. In fact they are reassuringly similar except in their perceptions of the likelihood of the potential for harm.

Editor-in-Chief: Unless you want to make a strong case for why we ought to consider harms that clinicians mostly do not regard as harms, it probably makes more sense to repackage your paper as an interesting exploration of how and why clinicians and patients (and members of the public) disagree in their characterisations of their care or scenarios depicting possible harm

There is a very strong case for this. It is at the very heart of patient trust and confidence in primary care. To use a quote by Berwick 1997 "The ultimate measure by which to judge the quality of a medical effort is whether it helps patients (and their families) as they see it. Anything done in health care that does not help a patient or family is, by definition, waste, whether or not the professions and their associations traditionally hallow it." The annual English GP patient Survey asks about trust and confidence in GPs because in the UK confidence and trust in primary healthcare is given a high priority. The next stage of this work is doing exactly as you suggest, we are exploring why there are such large differences in opinion but the data we collected in this work does not provide this information.

Editor-in-Chief: I am not devaluing the role of patients or members of the public in judging safety issues. But, we already have a substantial burden of harm documented from studies conducted by clinicians and/or involving clinician record review. So, there is no shortage of targets for future improvements – in test result communication, diagnostic accuracy, medication management, etc etc. There is even an exiting literature highlighting the role of patients as sources of information about patient safety. So, the conclusions about the frequency of harm as judged by patients are not at all the most interesting/novel part of your study, and these results are undermined by the finding that clinicians usually disagreed that the events involved harm.

So, it makes more sense to embrace the aspect of your study that compares clinicians' views and patients' views about these scenarios.

We have embraced this aspect of our study, it is one of our main findings. It is not so much about who judges safety but setting out the problem and showing that there is a wide difference in opinion. Surely clinicians would be interested to know when their patients feel “unsafe” so they can reassure them or rectify any genuine problems. We show that this patient-perception happens more often than clinicians might believe given that fewer than half of patients discuss their perception with a clinician. Furthermore we have also shown that there are probably no new “targets” from the patient perspective.

Some other specific comments

1. The title: please do not use GB as an abbreviation in the title. It's probably not worth including GB at all in the title. More generally, re-craft a title that highlights what the main aim of the study is in view of the above comments.

This has been done. A new title is suggested below.

“It all depends on your point of view; how do clinicians view patient-perceived potentially-harmful preventable-problems in primary care?”

2. “Of 3996 members of the public invited to participate, 3984 (99.7%) agreed to complete the questions relevant to this study and 3975 (99.5%) actually completed all the questions.” This is hard to believe. I have never heard of a survey in which virtually everyone approached agreed to participate. Was there some previous screening or outreach step that you are not reporting?

Author's response: All the participants had already agreed to participate in the Ipsos MORI face to face omnibus survey which occurs every week. To give an overview of the methodology - 170-180 sampling points are randomly selected to provide representation from all geographical locations. Interviewers within each sampling points are given quotas that are reflective of that sampling point and required to get the interviews from a pre-selected small group of streets. The interviewers then work through the streets in question and gain the interviews with various rules in place (e.g. can't interview in subsequent houses and can't complete their assignment on one street only etc.). As it is an approach that is random sampling at the offset, with quota sampling at a local level, response rates are not calculated as each interviewer is working on a quota sample basis and response rates can't be accurately determined. Typically an interviewer would get a completed interview from 1 in every 10 to 12 addresses attempted, however this would also include situations such as no one answers the door in the address and people being out of quota.

Some of the above information has been added to the methods section as shown below so that the reader is aware of this sampling procedure.

P4, lines 9 to 12: “Briefly 170-180 geographically representative sampling points were randomly selected and interviewers were required to get the interviews from a small group of streets reflecting that sampling point. (Typically an interviewer would get a completed interview from 1 in every 10 to 12 addresses.)”

3. The Abstract's Conclusion does not follow at all from the reported results.

Authors' response: We have changed the conclusion in light of this comment and the Associate Editor's comments. Our new conclusion is

“Addressing the differences in patient and clinician's views about preventable harm in primary care might improve patient trust and confidence in primary care.”

Further comments from the authors:

Reference 26 has been accepted for publication in BMJ Open and we have updated the manuscript to reflect this.

We have added a figure showing the distribution of the types of problem reported according to their route through the survey, i.e. originating from the first screening question or the prompt question, in Fig C in online Appendix 3. If the Editors do not agree with this addition we will remove it.

Due to addressing the reviewers' and Editors' comments the word count is now over the 4000 word limit. We are happy to revise the manuscript to reduce the length but for now we think it's important that the Editors and reviewers see that we have tried very hard to address their comments.

As the reviewers and Associate Editor have been largely supportive we hope that the Editor-in-Chief will recognise the value of the work in setting out the dilemma we face in reconciling clinician and patient views in relation to safety.

VERSION 2 – REVIEW

REVIEWER	Heyhoe, Jane Bradord Institute for Health Research, Quality and Safety
REVIEW RETURNED	24-Oct-2017

GENERAL COMMENTS	In general, the authors' revisions have resulted in a more focused and coherent paper and reports interesting findings with clear implications for research and practice. The method and results sections benefit from structuring the sections around the study aims. However, I found the "Summary of main results" repetitive in parts and wonder whether any additional information here should just be incorporated in the earlier sections of the results? In addition, I am not sure whether the aims of 'how the problem was discussed (if it was) and patient suggestions as to how it might have been prevented' should be separated from the other points in aim (ii). At the moment these interesting considerations seem to get a little lost among the other aims. I still found parts of the background and discussion a little disjointed and it is these sections that I feel would particularly benefit from further revision. For example, the second paragraph of the background could be re-organised so that it finishes with the description of the PREOS-PC and what gaps it might fill and how. While the discussion section includes clearer suggestions and reflections on implications, the content is still a little jumbled in parts. This might be helped by ensuring that the key findings for all aims are addressed before the "Strengths and weaknesses of the study" section. The point made in line 22-23 (p.11) "In the long term strategies to reconcile these differing perspectives need to be developed" - does the point need to be made that before strategies can be developed, we need to understand more about these different perspectives first? What type of work is required to do this? From Line 23 (p.11), "In the short term..." - this section may benefit from some re-wording. The final paragraph is a bit abrupt. Why is "the clinicians' perspective undoubtedly the most influential"? Perhaps reflect more on the importance of a shared understanding
--

	for improving safety? Some additional minor comments Page 11, Line 18 - no need for question mark after "public" Page 11 Line 22 - "do they have" not "had"?
--	--

VERSION 2 – AUTHOR RESPONSE

Reviewer(s)' Comments to Author:
Reviewer: 1

The authors would like to thank the reviewer for the helpful comments below.

Comments to the Author

In general, the authors' revisions have resulted in a more focused and coherent paper and reports interesting findings with clear implications for research and practice. The method and results sections benefit from structuring the sections around the study aims.

Author's response: Thank you.

However, I found the "Summary of main results" repetitive in parts and wonder whether any additional information here should just be incorporated in the earlier sections of the results?

Author's response: Thank you for this comment. We have done this by removing the summary and incorporating the extra points in to the main methods section. See p 6-9

In addition, I am not sure whether the aims of 'how the problem was discussed (if it was) and patient suggestions as to how it might have been prevented' should be separated from the other points in aim (ii). At the moment these interesting considerations seem to get a little lost among the other aims.

Author's response: Thank you for this very useful suggestion. We have done this. See highlighted text in the marked copy on p5, lines 2 to 3.

I still found parts of the background and discussion a little disjointed and it is these sections that I feel would particularly benefit from further revision. For example, the second paragraph of the background could be re-organised so that it finishes with the description of the PREOS-PC and what gaps it might fill and how.

Author's response: Thank you for this very useful suggestion. We have done this. The PREOS-PC has only reported on qualitative outcomes so far. However it is administered through primary care practices and therefore will not be able to report quantitative findings at the population level. It does, however, have the advantage for potentially linking to primary care records although I do not know if this is one of their aims. See the highlighted text in the marked copy on p4, lines 30 to 34.

While the discussion section includes clearer suggestions and reflections on implications, the content is still a little jumbled in parts. This might be helped by ensuring that the key findings for all aims are addressed before the "Strengths and weaknesses of the study" section.

Author's response: Thank you for this very useful suggestion. We have done this by moving the discussion of the findings that was under the strength and weaknesses heading further up the paper. See highlighted text in the marked copy on p10, lines 1 to 21.

The point made in line 22-23 (p.11) "In the long term strategies to reconcile these differing perspectives need to be developed" - does the point need to be made that before strategies can be developed, we need to understand more about these different perspectives first? What type of work is required to do this? From Line 23 (p.11),

Author's response: yes absolutely we should have made this point and have now done so. Thank you for pointing this out. Of course it is not an easy task to reconcile these views but so far preliminary

work has shown that members of the public are able to appreciate the clinician's perspective when it is explained to them. See highlighted text in the marked copy p11, lines 42 to 45.

"In the short term..." - this section may benefit from some re-wording. The final paragraph is a bit abrupt. Why is "the clinicians' perspective undoubtedly the most influential"? Perhaps reflect more on the importance of a shared understanding for improving safety?

Author's response: Thank you, we have removed this paragraph. We were basing this statement on the observation that the patient view is not prominent in the literature (arguably, depending on your perspective) and that clinicians are in a position to be more influential. Furthermore the comments by the Editor-in-Chief stated "Unless you want to make a strong case for why we ought to consider harms that clinicians mostly do not regard as harms, it probably makes more sense to repackage your paper as an interesting exploration of how and why clinicians and patients (and members of the public) disagree in their characterisations of their care or scenarios depicting possible harm" and we were trying, albeit unsuccessfully in the end, to refute that point of view. However perhaps it is not relevant and not evidence-based so it has been removed and, as you suggest, we have focussed on the importance of a shared understanding in our final statement.

Some additional minor comments

Page 11, Line 18 - no need for question mark after "public"

Page 11 Line 22 - "do they have" not "had"?

Author's response: Thank you very much for pointing these out and we have corrected them.

Response to rejection letter sent by the Deputy Editor in Chief of BMJ Q&S

Thank you for the opportunity to review your manuscript entitled "It all depends on your point of view; how do clinicians view patient-perceived potentially-harmful preventable-problems in primary care?" which you submitted to BMJ Quality & Safety. I am sorry to say that we have decided not to proceed further with the manuscript.

We have discussed the paper extensively in the editorial team, and it has been re-reviewed by the editor-in-chief as well as another senior editor and the handling editor. We all agreed that the paper had improved since the last draft, and we appreciated your detailed response to the previous reviews. We very much appreciated the quality of patient and public involvement in this study. However, some of the editors still had some serious concerns.

As noted by the reviewer (below – *now above*) the paper is still quite disjointed and hard to follow in parts,

Author's response: we have responded to the reviewer regarding this point above.

and would benefit from improved academic writing.

Author's response: we apologise for the poor quality of the writing. This paper contains a wealth of quantitative data that will be of use to those working in the field of quality and safety in primary care but we agree that it may be challenging for the casual reader. We have made further changes and tried very hard to guide the reader through a large amount of data and information.

The abstract fell considerably short of the standard we expect; it lacks clarity and coherence.

Author's response: We have changed the abstract to remove the part about the inter-rater agreement even though this was an important finding because it is difficult to summarise this concept in a few lines.

Though the title is about clinicians' views of patients' concerns, there is no mention of clinicians in the Methods section.

Author's response: The title had been changed in response to the Editor-in-Chief's comments and we have now returned to the original title which we feel is much more appropriate. In the methods section we state "Five GPs, one general dental practitioner and 7 members of the public estimated the likelihood that, in their opinion, each patient-described scenario was a potentially-harmful preventable problem.(24) The dental scenarios were only rated by the general dental practitioner and members of the public." Reference 24 describes in detail how the GPs rated the scenarios.

The abstract's conclusions concern trust and confidence, yet they are not mentioned as a focus of empirical concern in the objectives or methods.

Author's response: Trust and concern as measured by the English GP patient survey was used as a benchmarking question to confirm that our population was representative of the English/GB population in terms of this concept. It served its purpose in that respect. This is explained in the methods section p5, lines 23 to 25 and is highlighted in the marked copy. We reported on our observation that this patient attribute was more positive in patients who had discussed their problem with a primary care professional compared with those who had not. It was not a predefined hypothesis of our work but an interesting observation that we chose to highlight in our reporting. We believe it provides a rationale for making it easy for patients to discuss their problems.

In the manuscript itself we found that the aims of the study were not consistent with those of the title or the abstract,

Author's response: As we state above we have returned to our original title which is consistent with the aims of the study. We do not agree that the aims in the abstract are different to those in the study, as far as we can tell they are identical.

and that the objectives were far too many in number. One consequence of this is that the clarity of messaging we sought on the previous review is still not evident. We appreciated that you had added extra detail to the Methods section but we still found it hard to follow.

Author's response: There are several objectives and if our aim had been to have multiple publications that were (arguably) easier for the reader we could have done so. However as we state above we believe that these data should be kept together to make a coherent body of work for the serious reader. There are several messages and it is not possible to force the information in to simple summaries but we have done our best to comply with this requirement.

The Discussion section tries to make far too many points and does not adequately locate itself in the literature, instead making many proposals for improvements that may be difficult to enact or on which there are already literatures.

Author's response: Given that this is the first time that the frequency with which patients perceive potential harm at the population level (i.e. generalisable to the GB population) has been reported the next stage of the work will be to consider ways to address the need to reconcile the differing views. We are making suggestions that could be followed up, we are not particularly advocating any of these approaches but, rather, opening up the discussion. We would be happy to remove these suggestions if they prove a barrier to publication. It is usual to conclude a paper that outlines a problem with suggestions for how the problem could be resolved. We do not agree that we should refrain from mentioning a potential solution because it might be difficult to enact, for example due to a perceived lack of resources given that resource allocation is a political, not a scientific, decision and may change over time.

Your conclusion that "patient trust and confidence in primary care could be improved by addressing all patient-perceived potentially harmful problems" is problematic. First, it is not really clear what concepts of trust and confidence you are working with.

Author's response: We make no judgement on what "trust and confidence" means in this context. The question is a stable metric used in the English GP patient survey implying that it has a generally well understood meaning to the public and that is why we used it to test that our surveyed population was similar to the wider population with respect to this metric. We think that, whatever the precise

meaning, it is reasonable to consider it as a marker of quality in primary care. Defining precisely what it means to the individual patient would be a research question that we are not attempting to address.

Second, we are not sure about your strategy of "educating patients about their responsibilities" - do you mean before or after an incident has occurred? Is this something primary care can reasonably expect primary care professionals to do? We are also not sure that it is realistic to expect that differences of opinion can easily be reconciled, and your suggestion about informal routes for raising concerns, while a valuable one, may be difficult to action (and we would have expected to see further reference to the literature on this).

Author's response: See our response above. We absolutely agree that reconciling patient and clinician views will not be easy but any step in that direction is likely to have positive benefits to many patients. We certainly did not intend to imply that this should be solely the responsibility of primary care professionals although they clearly should be involved. While informal feedback might be considered resource intensive there may also be savings such as patients using primary care less frequently. We are talking about ongoing patient education, not in response to a particular concern. We agree that all of this needs debate and we simply do not have the space in this publication to discuss the literature in any detail. We are happy to remove all suggestions for addressing the problem and simply leave the paper as one that describes the current situation if that is deemed necessary for publication. However the main focus of our paper is to describe the current situation with respect to patient-perceived potential harm – the next stage of the work is to look more deeply at the underlying factors and propose strategies for reconciling these views.

Finally, we were a little alarmed at the changes in the conclusions and focus of the paper between the drafts, and felt that the paper overall continued to lack the coherence we expect.

Author's response: these changes were all made in an effort to satisfy the Editor in Chief's feedback. We agree that they were not changes for the better given that the reviewers were generally happy with the first draft of the paper. It would have been better if the Editor-in-Chief had rejected the paper in the first instance rather than suggest changes that were then used by the Deputy Editor-in-Chief as grounds for rejection.

I know this will be a very disappointing and frustrating outcome for you, especially given the effort you put into the revisions. I thus want to emphasise that this paper has had a lot of editorial attention, and also to give some context: we get over 1400 submissions a year, and can accept fewer than 10%. That means we have to make often difficult decisions about priorities. I am sorry we could not proceed with your paper in this instance. I hope the outcome of this specific submission will not discourage you from the submission of future manuscripts. We also hope you find an alternative journal for the paper in short order.

VERSION 3 – REVIEW

REVIEWER	Alisa Khan Boston Children's Hospital
REVIEW RETURNED	15-Feb-2018

GENERAL COMMENTS	Thank you for the opportunity to review this manuscript. The authors have tackled an important issue and have done a very nice job of engaging patient stakeholders as partners in designing and implementing the study. They have reviewed a great deal of data and conducted a large number of thoughtful analyses to reach their conclusions. I have some suggestions to improve cogency of the paper. Major: (1) My main feedback is that the manuscript goes a bit into the "weeds," and the reader is left somewhat missing the forest for the
--

trees by getting lost in sometimes distracting details that take away from what would otherwise be a paper with very profound findings. Overall, I might frame the findings a bit differently. I think the main takeaways of the authors' findings are that patients commonly recognize potentially harmful problems in primary care, that they have valuable insights for systems changes, but that they infrequently report these to their providers, suggesting our systems don't effectively engage families as safety partners. The fact that this patient recognition of problems does not necessary correlate with impaired trust is profound, and suggests even more strongly that healthcare systems should actively engage and solicit patient concerns more than they currently do--both in the inpatient and primary care setting. To me, this is the main argument that is more cogently supported by the authors' findings.

(2) I think the authors go into excessive detail, much of which is important and interesting, but that this distracts from the main message and importance of their findings. I would suggest they greatly simplify their presentation of results, allow the reader to interpret from the tables, rather than simply rehashing findings in the text. I would also suggest they streamline the discussion to make such an argument more focused and streamlined. In some ways, the paper is written more like a grant than an actual manuscript, which is I think part of the problem with its cogency (e.g., dividing methods by aim). I would also reduce the number of appendices included and referred to in the text as this is overwhelming and distracting to the reader.

(3) Things I found less interesting and more distracting were:

- The emphasis on the discrepancy between provider and clinician attribution of harm as an outcome (and a main conclusion). This discrepancy in the extent to which patients and clinicians recognize these problems as actually harmful may relate to how they define harm. I think this is interesting to note, but shouldn't be the main conclusion of the study, as the abstract currently frames it. I don't think getting them to agree is the holy grail; rather, I think this finding is interesting and simply needs to be better understood, but shouldn't be the focus.
- The non-leading screening questions discussion. (This seems tangential and I disagree with the premise. I would argue that if you frame what you are looking for up front, you will get better data.)
- That 69% of scenarios providing adequate info for clinicians to estimate (could this be removed or made simply a line in the limitations) (e.g., abstract and page 11). I didn't understand what this meant in practice. Why would there not be enough info to rank? What info is needed to rank?
- The details of the study specific aims - i think you can leave the objectives of the study, without going into the weeds of the specific aims as well. Including the specific aims may make the authors feel like they need to address all in this manuscript. They don't and in fact shouldn't. I would limit the manuscript to addressing only: a) rates of problems, predictors of problems (though deemphasize this), suggestions for preventing problems. If specific aims are included, they should be included in a much more concise, truncated fashion.
- Event rates per patient vs. consultations (page 10) - this is in the weeds. consider removing

(4) Page 7-8 (especially 7) of Results was very difficult to follow and overly detailed. Consider simplifying and simply picking a few key

	results and drawing reader's attention to the tables with the pertinent data. Subheadings may also help make the text easier to follow. Mainly, the text should be streamlined though. (5) Page 9 Discussion - the main finding "the proportion fell to 3%... 0.6%.." This is overly emphasized and to me not at all the most interesting finding of the study. Also the 8x more likely to lose confidence figure - is that mentioned in the Results? It should be because it seems odd that I first noticed it in the discussion. Again, I'm also not sure what "reconciling the patient and clinician perspective" means or that it is supposed to be the goal of this work. Abstract Edits -Results: add a parenthetical explanation of the f2f Omnibus for those not familiar with it -Results: As above, add the n/% of respondents discussing concerns with their -Results: Consider removing or simplifying the "24(0.6%) and 97 (2.4%) sentence." The 8% vs 39% is easier to digest (as in page 8 of results) - consider including this figure instead. -Results: remove sentence about "the strong emphasis.. problem." Didn't understand this or find it necessary. -Conclusions: Consider instead writing about how patient concerns should be actively solicited since patients frequently recognize concerns, have valuable suggestions for improvement, and reporting does not necessarily impede trust. Minor: (1) Define (briefly, even parenthetically) the NRLS and the f2f omnibus - I'm still not sure what the purpose of the latter in particular is (e.g., what topics does it cover?). (2) Methods, page 6 - Higher threshold and lower threshold -is confusing and overly detailed. In particular, I didn't follow what the "at least one score" was referring to (3) Add primary care throughout discussion, e.g., first line of discussion. It is important and noteworthy that the study was conducted in this population and should be emphasized in the discussion. It is mostly left out from it. (4)Page 10 - Ethnic minorities - rather than "important not to stereotype people" (not sure what this is getting at), perhaps say something like "providers need to meet the needs of all groups, including minorities, to speak up" (5) Page 11 - the wording suggesting patients need to have more realistic expectations of primary care is a bit awkward and seems to be blaming patients. Consider rephrasing in a more neutral manner.
--	---

REVIEWER	Dr Helen Hogan London School of Hygiene and Tropical Medicine, UK
REVIEW RETURNED	19-Feb-2018

GENERAL COMMENTS	This paper describes the findings of a population-level survey that set out to estimate the frequency of patient-perceived potentially harmful problems occurring in primary care. Its strengths include:  • a good sample size with an excellent response rate • a population-based survey rather than one undertaken with patients through particular healthcare providers • makes a valuable contribution to the understanding of the scale of
--

perceived potentially harmful problems amongst the public and the association of such problems with trust in the primary care provider

- reporting on problems per patient rather than per consultation better reflects the way problems might emerge along the patient journey and across a period of time in primary care
- use of an initial open question in the survey to allow respondents to report events that they felt might represent problems without the constraints of an imposed framework
- involvement of the public and patients in the design of the study
- comparison of ratings for patient-perceived potentially harmful problems between public and primary care provider reviewers
- provides a window on problems in care that arise from interactions with other primary care providers beyond GPs
- provides illumination of the type of problems experienced through case scenarios

I can see that the authors have already made a large number of changes to the manuscript following peer and editorial review which have improved its accessibility. Some additional areas to consider which might improve the paper further include:

As this is a cross-sectional survey, I think it should be stressed that it is not possible to say whether the occurrence of a problem leads to a loss of trust in affected patients or established low trust makes it more likely that a patient will perceive a problem has occurred.

Furthermore, those who have trust in their GP already may be more likely to discuss any problems that emerge, rather than the discussion itself having an impact on repairing damaged trust.

Given my point made above, I don't think it is right to say "respondents perceiving a safety problem were eight times more likely to lose confidence or trust in the GP". It is more accurate to say "that patients who perceived a problem were found to be eight times less likely to have trust in their GP"

Clinicians are less likely to identify mental distress as a consequence of a patient safety incident compared to patients. Can the authors present any data on the likelihood of a problem being perceived as probably or possibly causing harm by clinicians versus the public when it has physical or mental sequelae?

I think the following sentence needs some further thought:

"Rather, our results suggest that it may be beneficial to educate patients about their responsibilities as a patient and encourage them to have more realistic expectations of primary care."

Many of the respondent scenarios represent poor experience of care and I feel it would be reasonable for these people to have expected a better service- I don't think this chimes well with your suggestion that patients need to be encouraged to have more realistic expectations of primary care!

In the section of the Discussion related to the underlying differences in opinion between clinicians and members of the public reviewers on whether a potentially preventable harmful event might have occurred, it might be worth reflecting on findings from previous studies of patient-reported healthcare-related harm that have found patients often report back on poor experience rather than (often clinician defined) clinical safety problems and any literature related to the relationship between poor experience and safety.

In the Discussion section the authors might also want to put more emphasis on the likely impact of better communication between patients and clinicians in building trust, clarifying expectations, ensuring understanding and also encouraging patients to share any perceived problems

	Minor improvements Better to use full term for Great Britain with the first introduction of the GB abbreviation in the abstract and the main paper. The sentence at the end of the third paragraph of the results is confusing: “The combination of an open-ended question (Q2, Box 1) and prompt question (Q10, Box 1) prioritised sensitivity over specificity (as intended) given that 21% of the perceived problems (79/379) were excluded from the analysis, mainly because the perceived problem was not preventable or did not occur in primary care (Figure A, Appendix 3).” I think it should be clarified that the respondents themselves excluded some of the perceived problems when they rated these problems as not preventable in the questionnaire.
--	--

VERSION 3 – AUTHOR RESPONSE

Reviewer: 1

Reviewer Name: Alisa Khan

Institution and Country: Boston Children's Hospital Please state any competing interests: None

Please leave your comments for the authors below

Reviewer 1: Thank you for the opportunity to review this manuscript. The authors have tackled an important issue and have done a very nice job of engaging patient stakeholders as partners in designing and implementing the study. They have reviewed a great deal of data and conducted a large number of thoughtful analyses to reach their conclusions. I have some suggestions to improve cogency of the paper.

Author's response: Thank you

Major:

Reviewer 1: (1) My main feedback is that the manuscript goes a bit into the "weeds," and the reader is left somewhat missing the forest for the trees by getting lost in sometimes distracting details that take away from what would otherwise be a paper with very profound findings. Overall, I might frame the findings a bit differently. I think the main takeaways of the authors' findings are that patients commonly recognize potentially harmful problems in primary care, that they have valuable insights for systems changes, but that they infrequently report these to their providers, suggesting our systems don't effectively engage families as safety partners. The fact that this patient recognition of problems does not necessary correlate with impaired trust is profound, and suggests even more strongly that healthcare systems should actively engage and solicit patient concerns more than they currently do-- both in the inpatient and primary care setting. To me, this is the main argument that is more cogently supported by the authors' findings.

Author's response: Thank you for these insightful comments and we agree with the reviewer's summary of the findings and their implications. We have substantially reorganised the discussion and the points that the reviewer raises above are now included in the first paragraph of the discussion. Also the abstract has been altered in light of these comments. See abstract p1 & discussion p9-12.

Reviewer 1: (2) I think the authors go into excessive detail, much of which is important and interesting, but that this distracts from the main message and importance of their findings. I would suggest they greatly simplify their presentation of results, allow the reader to interpret from the tables, rather than simply rehashing findings in the text.

Author's response: We have removed much of the detail in the results section that repeated the information found in the tables, figures and appendices. We have also reduced the amount of information in the results and this is described in response to another point below.

I would also suggest they streamline the discussion to make such an argument more focused and streamlined.

Author's response: This has been addressed in the reviewer's first point above. See discussion p9-12.

In some ways, the paper is written more like a grant than an actual manuscript, which is I think part of the problem with its cogency (e.g., dividing methods by aim). I would also reduce the number of appendices included and referred to in the text as this is overwhelming and distracting to the reader.

Author's response: We agree that dividing the methods by aim was unhelpful and had been undertaken in response to a previous review by BMJ Q&S. We have removed the references to the aims in the methods. We have reduced the number of appendices to 2 by combining appendices 1- 3 in to a single appendix. The second appendix only includes the patient-described scenarios and the reader may follow the paper without any reference to this appendix although we think it is informative for the readers to be able to see the type of problem reported for themselves and makes the coding more transparent.

Reviewer 1: (3) Things I found less interesting and more distracting were:

-The emphasis on the discrepancy between provider and clinician attribution of harm as an outcome (and a main conclusion). This discrepancy in the extent to which patients and clinicians recognize these problems as actually harmful may relate to how they define harm. I think this is interesting to note, but shouldn't be the main conclusion of the study, as the abstract currently frames it. I don't think getting them to agree is the holy grail; rather, I think this finding is interesting and simply needs to be better understood, but shouldn't be the focus.

Author's response: We agree with the reviewer that this is only one aspect of the findings of this study and not one of the main findings. It has been demoted from the first paragraph of the discussion which is now more in line with the reviewer's observations in her first point above.

The previous review by BMJ Q&S had requested more emphasis on this aspect of the study. However we do think it is interesting plus the 2nd reviewer has focussed on this aspect of the study so we prefer to retain it in the discussion albeit with lower priority.

Even so we do believe that if members of the public and clinicians had a better understanding of each other's points of view it might improve safety and would very likely help to improve the public's perceptions of safety in primary care. However, as the reviewer points out, this is not the same as agreement between clinicians and members of the public. Our original aim was simply to ask clinicians to validate which problems were likely to be potentially harmful. However because the variation was so high we felt we could not ignore this large discrepancy in opinion both within clinicians and between clinicians and members of the public. The clinicians tended to disagree that the patient-perceived problems were likely to be potentially harmful and if this is communicated to the patient this may partly explain why patients feel uncomfortable about discussing said problems so we do feel it is relevant to the discussion about why patients are not discussing their perceived problems with primary care professionals.

-The non-leading screening questions discussion. (This seems tangential and I disagree with the premise. I would argue that if you frame what you are looking for up front, you will get better data.)

Author's response: This was only mentioned briefly in the methods and not discussed in detail. It was discussed in more detail in the methods paper (see <http://bmjopen.bmj.com/content/8/2/e017786>) but not in this manuscript.

-That 69% of scenarios providing adequate info for clinicians to estimate (could this be removed or made simply a line in the limitations) (e.g., abstract and page 11). I didn't understand what this meant in practice. Why would there not be enough info to rank? What info is needed to rank?

Author's response: This has been removed from the "strengths and weaknesses" section of the abstract. It was included because it could be a marker of the quality of the patient responses to the survey and possibly a more detailed interview by an appropriately trained person could have obtained the information needed to allow for these scenarios to be ranked by clinicians. We believe it should be discussed even if it is not sufficiently important to be listed as a specific weakness because for 31% of those reporting a problem we did not have a clinician estimate of the likelihood it was actually a problem. It is important for the transparency of the methods and interpretation of the results. It is an experimental limitation.

To answer the other questions (Why would there not be enough info to rank? What info is needed to rank?): Besides the quality of the information provided by the respondent, the individual ranker's (member of the public or clinician) varied in the level of evidence required before they were willing to offer an opinion. It reflects the individual's view on where the balance between sensitivity and

specificity should lie. The public tended to err on the side of caution (more sensitive) whereas clinicians preferred hard evidence before deeming it a potentially harmful problem (more specific). Alternatively the patients might be more inclined to trust the survey respondent's opinion than the clinicians. It is a complex interaction and we cannot really unpick it. As we say above our original aim was to use clinician judgement to validate the patient-reported problems but instead it gave us a different result.

-The details of the study specific aims - i think you can leave the objectives of the study, without going into the weeds of the specific aims as well. Including the specific aims may make the authors feel like they need to address all in this manuscript. They don't and in fact shouldn't. I would limit the manuscript to addressing only: a) rates of problems, predictors of problems (though deemphasize this), suggestions for preventing problems. If specific aims are included, they should be included in a much more concise, truncated fashion.

Author's response: We have truncated the aims and removed the numbering. We also demoted the clinician ranking to a secondary aim. However we do feel that it is important to specify our aims precisely and specifically so prefer not to cut back on this any further. See p4 line 42 to p5 line 5.

-Event rates per patient vs. consultations (page 10) - this is in the weeds. consider removing

Author's response: We think that this is an important difference between our study and other quantitative studies and prefer to keep it in the manuscript. Patients tend to view problems as evolving during their patient journey whereas clinicians tend to see a problem as an isolated event in a single consultation, the denominator is very different. This is discussed in Charles Vincent's book (reference 2) and because we discuss the use of time as a tool by clinicians we think this is quite important.

(4) Page 7-8 (especially 7) of Results was very difficult to follow and overly detailed. Consider simplifying and simply picking a few key results and drawing reader's attention to the tables with the pertinent data. Subheadings may also help make the text easier to follow. Mainly, the text should be streamlined though.

Author's response: We have removed some of the results from page 7. However many results are only shown in Appendix 1 and we are reluctant to completely remove any description of these results as we do refer to them in the discussion. The difficulty is that it is quite subjective to decide which results are more important. Every reader and reviewer has different idea about what is most important. If we report factually on all the results in this section then draw attention to what the reviewer and authors agree are the most important results in the discussion we think this is a good compromise. We do have 2 sub-headings in the results section.

(5) Page 9 Discussion - the main finding "the proportion fell to 3%... 0.6%.." This is overly emphasized and to me not at all the most interesting finding of the study.

Author's response: This has been removed from the discussion.

Also the 8x more likely to lose confidence figure - is that mentioned in the Results? It should be because it seems odd that I first noticed it in the discussion.

Author's response: The 8x more likely to lose confidence is mentioned in the results and the OR is shown in Table 1. See p7 lines 33-35

Again, I'm also not sure what "reconciling the patient and clinician perspective" means or that it is supposed to be the goal of this work.

Author's response: This was not a goal of this work but a poor choice of words for promoting a greater understanding of each other's point of view for patients and clinicians. It does imply we seek agreement which is probably impossible so we have just mentioned this in passing in the discussion and are happy to remove it if the reviewer or editor so wishes. See p12, lines 8-9 and pasted below.

"While reconciling the differing perspectives of patient and clinician may not be realisable, our study suggests that providing opportunities for, and encouraging, patients to discuss their concerns informally with a member of the primary care team may help with building trust, clarifying expectations and ensuring understanding."

Abstract Edits

-Results: add a parenthetical explanation of the f2f Omnibus for those not familiar with it

Author's response: Thank you for the suggestion. We have removed this from the abstract as it is not really possible to describe this survey which is defined by its methodology in a way that is informative but sufficiently brief as to comply with the abstract word limit. It is described in detail in the methods section. See p5, lines 11-23.

-Results: As above, add the n/% of respondents discussing concerns with their

Author's response: Thank you, this has been done. See p2 line 16.

-Results: Consider removing or simplifying the "24(0.6%) and 97 (2.4%) sentence." The 8% vs 39% is easier to digest (as in page 8 of results) - consider including this figure instead.

Author's response: Thank you, this has been removed.

-Results: remove sentence about "the strong emphasis.. problem." Didn't understand this or find it necessary.

Author's response: This was reported on because we thought that the study might reveal types of safety problem that were unique to the patient perspective, which it did not. We agree it is unnecessary in the abstract and has been removed.

-Conclusions: Consider instead writing about how patient concerns should be actively solicited since patients frequently recognize concerns, have valuable suggestions for improvement, and reporting does not necessarily impede trust.

Author's response: Thank you for the suggestion. It has been changed and now reads as below. See p2, lines 25-28.

"this study highlights the importance of actively soliciting patient's views about preventable harm in primary care as patients frequently perceive potentially-harmful preventable-problems and make useful suggestions for their prevention. Such engagement may also help to improve confidence and trust in primary care."

Minor:

(1) Define (briefly, even parenthetically) the NRLS and the f2f omnibus - I'm still not sure what the purpose of the latter in particular is (e.g., what topics does it cover?).

Author's response: Thank you for pointing this out. It has been done. See p5, lines 11-23 & p4, lines 22-25.

(2) Methods, page 6 - Higher threshold and lower threshold -is confusing and overly detailed. In particular, I didn't follow what the "at least one score" was referring to

Author's response: Thank you for raising this because it is important that the reader understands this logic if we are going to make any estimates of how many of the problems were likely to be potentially harmful in the public or clinician's opinion. We have rearranged this part of the methods to try and be clearer, see p6, lines 3-14. It can only be properly understood by referring to Appendix 1. It has also been described in reference 24.

"at least one score" has been changed to "at least one person gave a score of" see p6 lines 8-12

We do appreciate that the reviewer believes that the disagreement between clinicians, members of the public and patients is less important. Our original aim in soliciting clinician's views was to validate the patient-reported scenarios and we were not expecting to find that there were such large differences in opinion. We agree that this paper is primarily about the patient perspective but we cannot ignore the possibility that patients might be perceiving events as unsafe when they are not. Furthermore it is only natural that problems that are agreed to be safety problems by clinicians are more likely to be tackled. If clinicians do not believe that there is a safety problem there may be some resistance to addressing the patient-perceived problem. There may well be a different approach required to address patient-perceived safety problems that clinicians do not regard as real safety problems.

(3) Add primary care throughout discussion, e.g., first line of discussion. It is important and noteworthy that the study was conducted in this population and should be emphasized in the discussion. It is mostly left out from it.

Author's response: Thank you, this is very important and we have added "primary care" to the discussion in several places.

(4)Page 10 - Ethnic minorities - rather than "important not to stereotype people" (not sure what this is getting at), perhaps say something like "providers need to meet the needs of all groups, including minorities, to speak up"

Author's response: This is getting at the observation there are very small differences between patients perceiving or discussing a problem or not in terms of socio-demographics in contrast to some previous qualitative work. The difference in discussing problems between white and non-white groups is of borderline significance ($p=0.09$ which could be due to an underpowered study). By stereotyping we were referring to previous qualitative work that might have encouraged clinicians to view particular types of people as more likely to "complain" which might impact on how seriously the problem was viewed. We just want to emphasise that there is no justification for this type of thinking. Since we realise this very speculative we have removed the reference to stereotyping. Our point is that every type of person was equally likely to perceive a potentially-harmful preventable-problem. The sentence now reads as below.

"...it is important to consider each person's problem equally and encourage all groups, including minorities, to share their concerns." See p10, lines 21-22.

(5) Page 11 - the wording suggesting patients need to have more realistic expectations of primary care is a bit awkward and seems to be blaming patients. Consider rephrasing in a more neutral manner.

Author's response: yes we agree and this is also raised by reviewer 2 and has been addressed below.

Reviewer: 2

Reviewer Name: Dr Helen Hogan

Institution and Country: London School of Hygiene and Tropical Medicine, UK Please state any competing interests: None declared

Please leave your comments for the authors below

Reviewer 2: This paper describes the findings of a population-level survey that set out to estimate the frequency of patient-perceived potentially harmful problems occurring in primary care.

Its strengths include:

- a good sample size with an excellent response rate
- a population-based survey rather than one undertaken with patients through particular healthcare providers
- makes a valuable contribution to the understanding of the scale of perceived potentially harmful problems amongst the public and the association of such problems with trust in the primary care provider
- reporting on problems per patient rather than per consultation better reflects the way problems might emerge along the patient journey and across a period of time in primary care
- use of an initial open question in the survey to allow respondents to report events that they felt might represent problems without the constraints of an imposed framework
- involvement of the public and patients in the design of the study
- comparison of ratings for patient-perceived potentially harmful problems between public and primary care provider reviewers
- provides a window on problems in care that arise from interactions with other primary care providers beyond GPs

- provides illumination of the type of problems experienced through case scenarios

Author's response: Thank you

Reviewer 2: I can see that the authors have already made a large number of changes to the manuscript following peer and editorial review which have improved its accessibility. Some additional areas to consider which might improve the paper further include:

As this is a cross-sectional survey, I think it should be stressed that it is not possible to say whether the occurrence of a problem leads to a loss of trust in affected patients or established low trust makes it more likely that a patient will perceive a problem has occurred. Furthermore, those who have trust in their GP already may be more likely to discuss any problems that emerge, rather than the discussion itself having an impact on repairing damaged trust. Given my point made above, I don't think it is right to say "respondents perceiving a safety problem were eight times more likely to lose confidence or trust in the GP". It is more accurate to say "that patients who perceived a problem were found to be eight times less likely to have trust in their GP"

Author's response: Thank you for pointing this out. We agree that it is extremely important to interpret this observation correctly. We have replaced the sentence exactly as the reviewer recommends and it now reads as below.

Discussion: "This is important, not only because patients may be experiencing genuine safety problems, but also because respondents perceiving a potentially-harmful preventable-problem were found to be eight times less likely to have confidence and trust in their GP (Table 1)." See p9, lines24-26.

In the results it is described like this: "Those responding "no, not at all" to the question about trust and confidence in the GP (Q1 Box) were around eight times more likely to report a problem compared to those responding "yes, definitely"(Table 1)." See p7 lines 33-35.

Reviewer 2: Clinicians are less likely to identify mental distress as a consequence of a patient safety incident compared to patients. Can the authors present any data on the likelihood of a problem being perceived as probably or possibly causing harm by clinicians versus the public when it has physical or mental sequelae?

Author's response: This is a good question but we did not ask the right question to answer this. We asked about the nature of the problem that caused the perception of potential harm rather than the consequences of the potentially harmful problem. In some cases the potential consequences were described by the patient and sometimes the consequences were obvious, for example having the

wrong tooth pulled out. It is probably fair to say that for most cases the problem caused the patient distress with or without physical harm. If you were asking about the public and clinician views about the likelihood that patient-reported scenarios related to mental health issues are potentially unsafe then you can see in Table F in Appendix 1 that the members of the public were more likely to agree with the patient that mental health-related scenarios were potentially harmful compared with clinicians (when the data is dichotomised). A different approach to analysing the data showed this more clearly and it is an interesting observation but as it involves a different method of analysis to make quite a minor point, and this paper is already quite large, we decided not to include it here.

Reviewer 2: I think the following sentence needs some further thought:

“Rather, our results suggest that it may be beneficial to educate patients about their responsibilities as a patient and encourage them to have more realistic expectations of primary care.”

Many of the respondent scenarios represent poor experience of care and I feel it would be reasonable for these people to have expected a better service- I don't think this chimes well with your suggestion that patients need to be encouraged to have more realistic expectations of primary care!

Author's response: Yes thank you for raising this and we agree that we could have chosen our words better. We have replaced the phrase with “clarifying expectations” as recommended by reviewer 1 and this particular sentence has been removed from the manuscript. See p12, line 11.

We provided the details of the scenarios in Appendix 2 because each reader may take a different view about whether the patient's expectation was reasonable. However reasonable is not the same as realistic. Appendix 2 shows that many patients have higher expectations of primary care than is being delivered (although some of the scenarios were really quite trivial and we have expanded Appendix 2 to include examples of scenarios where all clinicians agreed that the scenario was definitely not “potentially harmful”). Often the patient's expectations are reasonable but, by definition, if they exceed the clinician's expectations they are probably unrealistic in terms of what the healthcare system can actually provide. Sometimes the patient may need to be proactive for their own safety. It is a case of being realistic about what is offered by primary healthcare which may differ to a patient's hopes or even reasonable expectations.

Reviewer 2: In the section of the Discussion related to the underlying differences in opinion between clinicians and members of the public reviewers on whether a potentially preventable harmful event might have occurred, it might be worth reflecting on findings from previous studies of patient-reported healthcare-related harm that have found patients often report back on poor experience rather than (often clinician defined) clinical safety problems and any literature related to the relationship between poor experience and safety.

Author's response: Yes we opened our introduction with this observation that patients are less likely to separate the concepts of safety and quality than clinicians.

“Patients and clinicians view safety differently; patients tend to consider both serious safety problems as well as lesser causes of distress as safety concerns.(1)”

We have also cited reference 1 in the discussion because it qualitatively analyses the differences in views of patient safety among the public and clinicians. It is consistent with our study but based on the interviews of just 38 patients. A quantitative paper (reference 41, Lawton et al) found no relationship between patient experience (measured by the friends and family test) and measures of patient safety from the patient or clinician perspective in UK secondary care. Conversely a Canadian paper (Kemp et al <http://bmjopen.bmj.com/content/6/7/e011242>) found that inpatient experience ratings were associated with patient safety indicators. We agree that this is an important question but to look at this question properly requires a proper literature review and it's probably beyond the scope of this paper which already includes 44 references.

Reviewer 2: In the Discussion section the authors might also want to put more emphasis on the likely impact of better communication between patients and clinicians in building trust, clarifying expectations, ensuring understanding and also encouraging patients to share any perceived problems

Author's response: Thank you and reviewer 1 made similar comments. Yes absolutely we should have done this and have concluded our discussion with these points. See p12 lines 8-11.

We also made this the conclusion of the abstract.

Reviewer 2: Minor improvements

Better to use full term for Great Britain with the first introduction of the GB abbreviation in the abstract and the main paper.

Author's response: Thank you for the suggestion, this has been done

The sentence at the end of the third paragraph of the results is confusing:

"The combination of an open-ended question (Q2, Box 1) and prompt question (Q10, Box 1) prioritised sensitivity over specificity (as intended) given that 21% of the perceived problems (79/379) were excluded from the analysis, mainly because the perceived problem was not preventable or did not occur in primary care (Figure A, Appendix 3)."

I think it should be clarified that the respondents themselves excluded some of the perceived problems when they rated these problems as not preventable in the questionnaire.

Author's response: Thank you for this helpful suggestion and this has been clarified. See p7, lines17-21 and pasted below.

"The combination of an open-ended question (Q2, Box 1) and prompt question (Q10, Box 1) prioritised sensitivity over specificity (as intended) given that 21% of the reported problems (79/379) were excluded from being a potentially-harmful preventable-problem in primary care by the

respondent themselves by their response to questions 4 and 6 (i.e. not preventable or not in primary care, Box1).”

VERSION 4 – REVIEW

REVIEWER	Alisa Khan, MD, MPH Boston Children’s Hospital, United States
REVIEW RETURNED	11-Apr-2018

GENERAL COMMENTS	Thank you for the authors’ thoughtful revisions. The framing of the discussion in particular is a marked improvement.
---

REVIEWER	Helen Hogan London School of Hygiene and Tropical Medicine, London, UK
REVIEW RETURNED	02-Apr-2018

GENERAL COMMENTS	I feel that the authors have responded adequately to the reviewer suggestions and as a result the paper has been improved and now reads much better.
--

VERSION 4 – AUTHOR RESPONSE

Reviewer(s)' Comments to Author:

Reviewer: 1

Reviewer Name: Alisa Khan, MD, MPH

Institution and Country: Boston Children’s Hospital, United States Please state any competing interests: None declared

Please leave your comments for the authors below Thank you for the authors’ thoughtful revisions. The framing of the discussion in particular is a marked improvement.

Authors’ response: Thank you for your helpful comments.

Reviewer: 2

Reviewer Name: Helen Hogan

Institution and Country: London School of Hygiene and Tropical Medicine, London, UK Please state any competing interests: None declared

Please leave your comments for the authors below I feel that the authors have responded adequately to the reviewer suggestions and as a result the paper has been improved and now reads much better.

Authors' response: Thank you for your helpful comments.